# Modelling regional glacier length changes over the last millennium using the Open Global Glacier Model

David Parkes[1] and Hugues Goosse[1]

[1]Earth and Life Institute, Universite Catholique de Louvain

**Correspondence:** David Parkes (david.parkes.88@gmail.com)

**Abstract.** A large majority of the direct observational record for glacier changes falls within the industrial period, from the 19th century onward, associated with global glacier retreat. Given this availability of data, and significant focus in contemporary glacier modelling falling on recent retreat, glacier models are typically calibrated using - and validated with - only observations of glaciers which are considerably out of equilibrium. In order to develop a broader picture of the skill of one glacier model - the Open Global Glacier Model (OGGM) - we model glaciers for extended historical timescales of 850-2004 CE using a selection of 6 general circulation model (GCM) outputs. We select glaciers for which long term length observations are available in order to compare these observations with the model results, and find glaciers with such observations in almost all glacierised regions globally. In many regions, the mean modelled glacier changes are consistent with observations, with recent observed retreat in these regions typically at the steeper end of the range of modelled retreats. However, on the scale of individual glaciers, performance of the model is worse, with overall correlation between observed and modelled retreat weak for all of the GCM datasets used to force the model. We also model the same set of glaciers using modified climate timeseries from each of the 6 GCMs that keep temperature or precipitation constant, testing the impact of each individually. Temperature typically explains considerably more variance in glacier lengths than precipitation, but results suggest that the interaction between the two is also significant within OGGM and neither can be seen as a simple proxy for glacier length changes. OGGM proves capable at reproducing recent observational trends on at least a qualitative level in many regions, with a modelling period over a considerably larger timescale than it is calibrated for. Prospects are good for more widespread use of OGGM for timescales extending to the pre-industrial, where glaciers are typically larger and experience less rapid (and less globally consistent) geometry changes, but additional calibration will be required in order to have confidence in the magnitude of modelled changes, particularly on the scale of individual glaciers.

## 1 Introduction

Robust modelling of the evolution of glacier mass and geometry on regional and global scales is of critical importance for understanding the components of historical sea level rise and for predicting one of the potential largest contributors to sea level rise in coming centuries (Church et al., 2013). Moreover, glacier geometry changes are themselves an indicator of local (for individual glaciers) and regional (for glaciers considered across a wider area) climate changes (Oerlemans, 1986, 2005). Direct observations of historical glacier geometry (observations of contemporaneous glacier extent, as opposed to secondary sources

like moraines or lake sediments) are relatively sparse (Zemp et al., 2015; Cogley, 2009), and it is only through recent aerial (WGMS and NSIDC, 1989) and satellite mapping (RGI Consortium, 2017) that fairly comprehensive inventories of glaciers across all of the world's glacierised regions have become available, cataloguing upwards of 200,000 glaciers. Even this number is likely a significant underestimate (Parkes and Marzeion, 2018). Investigating the past and future state of glaciers on large scales therefore necessitates the use of glacier models that can accurately forecast or hindcast glacier evolution with relatively little historical observational data for calibration, and without a prohibitive computational cost. This is one of the primary goals of the Open Global Glacier Model (Maussion et al., 2019) project.

With much of the focus of glacier modelling on recent retreat and future predictions, a longer term historical view of glacier changes provides valuable context. For glacier models developed using data primarily collected in the 20th and 21st centuries, it is important that these models are examined over time periods where more stable glacier geometries were expected. OGGM by default calibrates the glacier sensitivity to local temperature based on CRU TS 4.01 data (Harris et al., 2020) that begins in 1901, and uses RGI version 6.0 (RGI Consortium, 2017) glacier outlines which are typically from around the start of the 21st century. In many regions, glaciers are already experiencing significant retreat by the beginning of the 20th century (Zemp et al., 2015), and calibrating glacier models for time periods when glaciers are far from equilibrium brings additional challenges. It is a critical test of a model's ability to determine whether it can reach non-trivial equilibrium states in periods of more stable climate, because recent retreats and expected future sustained retreats necessarily exist as transitional phenomena between equilibria. In the case of OGGM specifically, the calibration of surface mass balance sensitivity assumes the model's ability to maintain an equilibrium geometry.

Studies suggest that the last millennium provides smaller - and less globally consistent - temperature trends than the industrial period record alone (Neukom et al., 2019; PAGES 2k Consortium, 2013, 2017). While there are discernible large scale temperature trends within the last millennium, notably the Medieval Climate Anomaly - c. 10th to 13th centuries CE - and Little Ice Age - c. 16th to 19th centuries CE (Neukom et al., 2019), none have either the spatio-temporal consistency or the magnitude of recent industrial warming. Modelling over a longer period incorporating both recent warming and earlier climates allows OGGM to be tested over a period where glaciers are expected to remain more stable, potentially with both advances and retreats at more moderate rates than have been recently observed. The small number of available length records which extend back further than 150-200 years (Leclercq et al., 2014; Solomina et al., 2016) heavily limits any possible comparison of model results with observed pre-industrial glaciers lengths, so a greater focus is placed on whether (and when) the modelled glaciers transition from relatively stable pre-industrial lengths to expected recent retreat than on pre-industrial trends and variability. The timing of the onset of modern glacier retreat exhibits complexity that is unlikely to be explained by only temperature and precipitation changes (Painter et al., 2014; Luthi, 2014; Sigl et al., 2018) so it is unlikely OGGM will be able to replicate the timing exactly, but we can still usefully compare the speed and magnitude of modelled recent retreat with observed retreat.

Another issue with large-scale glacier modelling is the highly spatially inhomogenous nature of historical records of glacier change. Observations are typically concentrated in more accessible regions, rather than distributed evenly by glacierised area, and this is compounded in the longest-term observations. This naturally biases glacier model development towards representation of certain regions - most notably central Europe - which do not contain a large proportion of global glacier mass, and

which are not necessarily representative of glaciers in larger regions (Hock et al., 2019). For this reason it is important to assess the performance of glacier models on a per-region basis, and to determine how well models can reconstruct observed changes in the best-observed regions compared less well-observed regions.

From another angle, as glaciers aggregate changes in local climate, modelling of glaciers under a modelled or reconstructed climate can be considered a test of the climate model/reconstruction's skill at reproducing the variables which drive glacier models (in the case of OGGM, temperature and precipitation). Usage of glacier models for this purpose is dependent on both the level of confidence in the glacier model's skill and the available observations to compare with modelled glacier states. There is potential for longer-term historical glacier modelling to provide a link between proxies for glacier extent, such as morraine positions and sediment deposition, and models or reconstructions of past climate (e.g. Daigle and Kaufman, 2009).

In this paper, we use OGGM to model a selection of glaciers for which long-term length observations exist (Leclercq et al., 2014) in almost all major glacierised regions over the last millennium, and compare modelled length changes to observations with the primary aim of assessing OGGM's performance over longer time periods and a diverse set of regions. Secondary goals are establishing the relative importance of precipitation and temperature forcing, and comparing the impact of forcing the glacier model with different GCM datasets. Driving OGGM with an ensemble of GCM output timeseries allows us to determine whether OGGM's results are robust when using a reasonable range of climate forcings. This approach has already been applied to European glaciers (Goosse et al., 2018) on a millennial timescale, and the extension to many glacierised regions adds valuable understanding of model behaviour and glacier dynamics in different climate conditions - including for future applications modelling entire global glacier inventories - and allows comparisons of differences between runs driven by diferent GCMs and of differences between regions. We also use modified climate variable timeseries with constant precipitation or temperature (including only high-frequency variation) for each of the climate models. The constant precipitation and temperature runs inform the sensitivity of modelled glaciers to each form of forcing individually, and serve the primary goal of model assessment by showing which forcings dominate and where this can be associated with OGGM's performance in reproducing observed glacier lengths.

## 2  Data description

### 2.1  Climate models

Our OGGM runs use input from 6 different climate models, each covering a period from 850 to 2005 CE. These models are CESM, IPSL, GISS, BCC-CSM, CCSM4, and MPI - using simulations under the Past Model Intercomparison Project (PMIP3) and the Coupled Model Intercomparison Project (CMIP5) protocols (Schmidt et al., 2011; Taylor et al., 2012; PAGES 2k-PMIP3 group, 2015) - with details exactly as listed in Goosse et al. (2018) table 1 and section 2.1, with the exception that only a single simulation from CESM is used. OGGM results are produced for the years 851-2004 CE, due to the requirement of clipping the data to match hydrological years. The GCMs provide gridded monthly records of temperature and total precipitation without a reference height, as OGGM calculates the actual climate timeseries used by taking the GCM anomalies compared to the 1961-1990 mean for temperature and precipitation and applying these to the CRU TS 4.01 (Harris et al., 2020)

1961-1990 means. The use of anomalies relative to the CRU data accounts for overall bias in the GCMs, and is part of OGGM's default processing for non-CRU datasets.

## 2.2 Glacier observations

Initial glacier outlines and topography are taken from the Randolph Glacier Inventory version 6.0 (RGI Consortium, 2017) and several digital elevation models (DEMs) respectively, using OGGM's default preprocessing at level 3. The DEMs used are SRTM (CGIAR-CSI, 2019), GIMP (NSIDC, 2019), and Viewfinder Panoramas DEM3 (de Ferranti, 2019), each for different regions. OGGM's level 3 preprocessing contains the outputs for all steps of the preparation of initial glacier state using default parameters, so the runs performed in this study can focus exclusively on running the dynamic glacier model in response to varying climate datasets with consistent parameters and initial glacier geometry.

In order to compare modelled length changes with a set of observations which covers most RGI regions, the glaciers we model are those featured in a 2014 dataset of observed glacier length fluctuations compiled by Leclercq et al. (2014). The identification of glaciers from the Leclercq dataset with glaciers in the RGI - necessary for modelling in OGGM - is non-trivial, and is described below. Length change observations are arguably the simplest metric of glacier geometry change as they can be determined using only snapshots of terminus location, which is why this observational dataset goes further back in time for many glaciers than reliable observations of glacier area or volume. The dataset shows certain biases which impact how well we can expect the glaciers to be globally or regionally representative samples. Firstly, the number of glaciers observed per region is not representative of either total glacier number or total glacier area (see Table 1 for regional totals used). Most notably, the Central Europe region has the largest number of observations, despite containing much less total glacier area than many other regions. This precludes the production of meaningful globally averaged figures in this paper. Secondly, larger glaciers are heavily over-represented in the Leclercq dataset compared to comprehensive modern inventories like the RGI (see figure 1), and larger glaciers may also still be overrepresented in the RGI (Parkes and Marzeion, 2018). This means that the response time of the Leclercq dataset glaciers may be expected to be longer than for glaciers in each region as a whole. Studies suggest that it is not glacier size itself that is a primary determinant of response time but glacier thickness (Jóhannesson et al., 1989; Harrison et al., 2001) - though thickness scales to an extent with glacier length and area (Bahr et al., 1997) - with an influence from slope, elevation range, and mass balance gradient (Zekollari et al., 2020). Whether response time actually increases with glacier size can also vary by region (Raper and Braithwaite, 2009). The comparison between model results and observations is not affected by response time differences directly, as all comparisons are like-for-like on specific glaciers or sets of glaciers, but it does mean that the changes shown are likely a) slower and b) smaller in relative magnitude compared to the true regional averages, which contain many smaller glaciers that typically have faster and proportionally greater responses to changes in local climate.

The length change timeseries in the Leclercq dataset vary considerably in the number of years covered and in the frequency of observations, with figure 2 showing the number of glaciers which have observed lengths available in each region. Within any given region, the number of available data points varies from year to year so it is important to choose a representation of mean regional glacier length that can handle this. All possible solutions have positives and negatives, but we opt to normalise

each glacier's length relative to length in a reference year when used in regional means so that it is possible to focus more on changes over time changes than on net biases between models. The reference year for normalisation is 1950, which is a requirement imposed by the Leclercq dataset as it is the year guaranteed to be covered by the timeseries for all included glaciers. For each modelled (or observed) glacier length timeseries, the normalised length timeseries is thus given by the length in each year divided by the length in 1950. Regional mean glacier length is calculated as the mean of this normalised length across all glaciers which have observations covering a given year. This reduces the 'spikes' in mean regional length which arise from changes in the number of glaciers with measurements in a particular year, especially when the glaciers joining or leaving the mean are far from the mean absolute glacier length. However, we do still see spikes in the Leclercq dataset averages as artefacts of the sampling, particularly in the earlier parts of the regional timeseries where the glacier number contributing to the average changes while the total number of glaciers is small. Normalised regional means remove the ability to immediately judge differences in absolute glacier length between models - indicative of the relative biases of the model overall, and of the ability of the bias correction technique to remove those biases - but makes comparing periods of advance and retreat between models much easier. Additional diagnostics, evaluating the absolute changes for individual glaciers, are also provided to complement the aggregated regional analyses.

Glaciers from the Leclercq dataset are identified with the glaciers in the RGI in two steps. First an attempt is made to find a positive match in the RGI for the glacier described in the Leclercq dataset, according to an objective standard, and if this fails, an attempt is made to find a nearby glacier which may not be confidently identified as the glacier described in the Leclercq dataset, but can be used as a 'best effort' for the purpose of comparing local glacier changes. These two types of identification are kept distinct, and labelled as such in the glacier list. To find positive matches, the criteria are the following: 1) the (lat, lon) pair given in the Leclercq data must either lie within the outline of an RGI glacier, or within rounding distance for the (lat, lon) values (which are given to 2 decimal places); and 2) the area given in the Leclercq data must be within a factor of 2 of the RGI glacier. In cases where the (lat, lon) pair given is exactly on the border between connected glaciers, or within rounding distance for more than one glacier but within the outline of neither, and both glaciers satisfy the 2nd condition, one glacier is selected but moved to the 'best effort' class (though occasionally this will not be necessary if one of the RGI glaciers can be uniquely identified with a different Leclercq glacier, as the other can then be positively identified as the correct Leclercq glacier). The 1st criterion is not applied as strictly in certain cases where the (lat, lon) pair is close to a larger glacier and there are no other glaciers nearby, particularly when the (lat, lon) location is clearly downstream of an RGI glacier's tongue as this represents a rapid tongue retreat between the time of the Leclercq measurement and the time of the RGI measurement. The 'best effort' criteria are much looser, simply selecting the most size-appropriate glacier in the local group of glaciers (but not any RGI glacier which is positively identified as a different Leclercq glacier). If there are either no local RGI glaciers or the given size of the Leclercq glacier is vastly different to that of any local RGI glaciers, no 'best effort' glacier is identified and the glacier status is given as 'not found'. For the 471 glaciers in the Leclercq dataset this process gives 291 positive matches, 121 best-effort matches, 38 not found, and the remaining 21 Antarctic glaciers unprocessed due to lack of CRU data for calibrating sensitivity in surface mass balance calibrations. 412 glaciers - the positive matches and best-effort matches - are therefore used for modelling.

## 3 OGGM and experiment description

### 3.1 Open Global Glacier Model

The Open Global Glacier Model (OGGM) version 1.1 (Maussion et al., 2019), updated to the most recent code version as of 2019-03-28, is used for modelling the glaciers in this study. OGGM is an open source model of glacier evolution, which couples a surface mass balance (SMB) model based on precipitation and temperature with a model of glacier dynamics. Glacier mass balance is calculated using a temperature-indexed degree-month model; monthly temperatures above a melt threshold drive ablation and monthly solid precipitation, calculated as a fraction of total precipitation based on a threshold temperature, drives

accumulation. This generates monthly surface mass balance at each specific elevation on the glacier surface. OGGM takes gridded monthly records of temperature and total precipitation from the GCM datasets, which it uses to determine temperature and precipitation at each specific glacier location by applying these as anomalies to the CRU TS 4.01 (Harris et al., 2020) mean climate from 1961-1990 at the CRU grid reference height, then scaling these to the glacier surface at each OGGM grid point using a default temperature lapse rate of -6.5°C / 1000m from the CRU reference height and a uniform precipitation multiplier

of 2.5 to account for enhanced precipitation in mountainous topography. Full details are given in the paper that describes the model (Maussion et al., 2019), and further background on the mass balance calculation is available in the precursor to OGGM described in Marzeion et al. (2012).

   OGGM calibrates the sensitivity to temperature ($\mu*$), which linearly scales the melt each month with temperatures above the melt threshold, using CRU TS 4.01 temperature and precipitation data for the 20th century (Harris et al., 2020) and WGMS

observations of glacier surface mass balance (WGMS, 2019). For glaciers where SMB data is available, the calibration process takes each year ($t$) of climate data in the CRU dataset as a candidate and determines a sensitivity value ($\mu$) in that year for which the net mass balance of the glacier in its current geometry would be zero (essentially assuming the modern glacier would be in equilibrium under that year's climate). The year $t*$, and corresponding sensitivity $\mu*$, is then chosen as the year for which the sensitivity best reproduces the observed SMB. For the majority of glaciers that do not have available SMB data, the year $t*$ is

interpolated from other glaciers based on proximity because the interpolation based on $t*$ gives better results than interpolating $\mu*$ itself. Again, full details of this process are described in Marzeion et al. (2012).

   Glacier length in OGGM, which is the primary output metric for this paper, is the along-flowline distance from the head of the glacier to the terminus along the major centreline, calculated using an implementation of the method from Kienholz et al. (2014). Each glacier can have multiple tributary flowlines. With the input of surface mass balance calculated as described

above, a flux-based ice dynamics model determines the evolution of glacier thickness at each coordinate along each flowline that comprises the glacier. The terminus of the glacier is sensitive to this thickness evolution: with positive flux 'overflow' from the final coordinate along the main flowline, the glacier will grow to the next coordinate, and if the thickness at the final coordinate reduces to zero, the new terminus will be the previous coordinate.

## 3.2 Experiment-specific setup

Two sets of runs are performed: a primary set of runs under 'full' climate forcing using the data from each GCM, and a secondary set of 'constant temperature'/'constant precipitation' runs. These secondary runs have the temperature/precipitation respectively in each year randomly selected from a year in the 1400-1450 (inclusive) window from each GCM's output, with the precipitation/temperature respectively the same as in the primary runs. The randomised 51-year window provides a temperature/precipitation time series that represents a constant long-term climate while preserving some degree of interannual variation, so that the impact of a lack of long-term trend in the temperature/precipitation values can be examined. The period of 1400-1450 is chosen for centrality within the dataset, and because it falls neither within the Medieval Climate Anomaly nor the Little Ice Age.

For all runs, a 300-year spinup using annual climate data selected randomly from a 51-year window of 875-925 CE from the same GCM is performed prior to the run, to allow the glacier to develop from the preprocessed glacier geometry (based on RGI data and therefore representative of the year of the glacier outline observation used). While this does not necessarily give a more realistic starting glacier geometry for the run, it does allow the starting geometry to get closer to equilibrium for the climate in the early part of the model run. If the starting glacier is not allowed to adapt to a start-of-run equilibrium, it is more likely a major adjustment will occur in the early part of the run due to the effective step-change in climate variables that forcing a 20th century glacier geometry with 9th century climate data would represent, particularly if the glacier's local climate at the start of the run and the climate in the late 20th century are very different.

All 412 of the glaciers matched between the Leclercq dataset and the RGI are modelled, but we cut down the results to exclude marine- and lake-terminating glaciers. Before any modelling concerns, the loss of ice at the glacier terminus through calving makes the terminus location (and therefore glacier length) a less useful indicator of glacier geometry changes, as the terminus can remain in a similar location through considerable thinning or thickening of the glacier while the calving flux changes. In the context of OGGM, the default settings used here do not include a parameterisation of calving, which has a large impact on glacier geometry. OGGM does allow for a parameterisation of calving flux (Recinos et al., 2018), but this still relies on a fixed location of the calving front that enforces a physically unrealistic calving-front thinning followed by a transition to a non-calving regime if the glacier is expected to retreat, so it does not improve our ability to examine the evolution of glacier geometry through the lens of accurately modelled length changes. 91 glaciers are excluded from the regional averages due to being marine- or lake-terminating, leaving 321 from which regional averages are determined. The numbers excluded per region are shown in Table 1, along with the number of glaciers contributing to the means for each region after failures in the modelling process.

## 4 Results

Figure 3 shows the regional mean length results for the years 1000-2004 CE for the runs using temperature and precipitation data as provided by each of the 6 climate models. The model itself runs using this data from 851-2004, but we limit our graphs to 1000 CE onwards in order to limit the impact of a continued adjustment towards equilibrium even after the 300 year spinup

for certain model/region combinations, which is most likely a modelling issue and not a response to actual climate trends. In cases where glaciers are still undergoing significant adjustment to a new equilibrium (e.g. in South Asia West) after several hundred years of spin-up and the early part of the main run, this is good evidence that in a 1000 year period, responses to trends in the forcing climate variables may not actually be shown in the OGGM output. This does not invalidate the glacier model output, but the evidence of continuing adjustment leftover from the spin-up being shown in the output rather than being removed with an arbitrarily long spin-up might inform the interpretation of the rest of the timeseries. It is also the case that where continued adjustment is significant after several hundred years, the magnitude of the length changes is typically large, and in these cases adding additional spin-up centuries will not fix the fact that the modelled glacier is diverging from the size of the observed glacier. We choose to maintain the 300-year spin-up for the sake of consistency as well as these reasons.

Supplementary material figures S11-S16 show the spread of all individual timeseries that contribute to each of the model means shown in figure 3. Table 1 shows the number of glaciers per region contributing to the mean for each of the climate models, with glaciers removed if they are not land-terminating or if there are modelling errors. Figure 4 also shows regional mean lengths, but unlike figure 3 the number of glaciers in each of the mean timeseries changes over time to match the exact set of glaciers included in the Leclercq mean for that year. This is limited to showing only model results from the first year in each region with available observations.

In order to determine the significance of the apparent retreat in the last ~150 years for many model runs, a 'split regression' is performed for each climate model in each region. For each year from 901 to 1954 (we remove the first and last 50 years of the timeseries to avoid looking for trends in timeseries which are small compared to the expected response times of glaciers), we split the model output into a section up to and including that year and a section after that year, and perform a simple linear regression on each part. The 'best' split - meaning the split which most effectively splits the timeseries into two linear trends - is chosen by maximising the summed $r^2$ values for the two sections. These best splits are shown in Fig. 5, and demonstrate that according to an objective standard for determining the separation of trends there is in many cases a clear industrial retreat. In 6 regions - Western Canada/US, Greenland Periphery, Central Europe, Low Latitudes, Southern Andes, and New Zealand - the runs for all 6 climate models show a distinction between pronounced recent retreat and a modest pre-industrial advance. In a further 7 regions - Alaska, Iceland, Svalbard, Scandinavia, North Asia, Caucasus/Middle East, and South East Asia - multiple climate models show a distinction between pronounced recent retreat and a pre-industrial trend (though in these cases the pre-industrial trend varies from moderate advance to moderate retreat). While it is not necessarily useful to consider the year of transition between the two regressions as the year there is a point of inflection in the glacier length, due to the restrictions imposed by separating into just two linear regressions, it is notable that in many cases, there is more variability in the year of the split than in the slope of the post-split retreat (the best examples of this are Southern Andes, Low Latitudes, and New Zealand). This suggests that differences in total industrial retreat are more influenced by differences in when the retreat starts than by differences in the severity of the retreat.

Metrics to compare modelled and observed glacier lengths which are not normalised or aggregated by region are shown in figures 6 and 7. Figure 6 shows the distribution of absolute differences between modelled and observed per-glacier lengths in 1950 for OGGM driven by each of the GCMs. We see a general underestimation of 1950 glaciers lengths forcing OGGM with

CESM data, a general overstimation of 1950 lengths using GISS, and a greater spread of length errors using BCC-CSM. For all models the whole of the interquartile range lies within the -2km to +2km range, with the context of the distribution of 1950 observed lengths in figure 10.

Figure 7 is a plot of 20th century length change trend for modelled glaciers vs observed glaciers for model runs driven by each GCM. The trend is calculated for all glaciers which have at least 68 years within the 20th century covered by the Leclercq observations, with 68 years chosen to allow over 90% of glaciers to be considered (289 of the 319 which contribute to regional averages). A linear regression of the scatter of trends is performed and shown on each graph, with the slope significantly less than 1:1 for all of the GCMs. In particular, for CESM this regression is essentially flat, indicating no globally coherent skill in

reproducing 20th century length changes for individual glaciers.

To understand the impact of temperature and precipitation individually in driving trends in the modelled output, we plot the primary output alongside the output with constant precipitation/constant temperature. Figures 8 and 9 show these results for each region for one climate model - IPSL - along with the smoothed annual precipitation/temperature (given in degree-days), with the same figures for the other 5 climate models appearing in supplementary material (Fig. S1-S10). IPSL is singled out

only for illustrative purposes; 20th century IPSL slopes are typically on the steeper end (and thus in many regions closer to observations), suggesting that differences between the full forcing run and the constant temperature and/or precipitation runs may be more visible in this period. Figure 10 also shows the variance in the constant precipitation/constant temperature runs relative to the variance in the primary runs.

Figures 8, 9, and 10 show that for most models and for most regions, temperature influences length fluctuations more

than precipitation, consistent with similar multi-regional studies on sensitivity to climate variables (Oerlemans and Reichert, 2000; Sicart et al., 2008), but the relative importance of the two factors is far from homogenous. In addition to particular GCMs which show anomalous relative importance of temperature and precipitation compared to other GCMs in a region - for example, CCSM4 in Iceland - there are also regions where the influences of temperature and precipitation are much more equal across most climate models, most notably South Asia West and South Asia East. The information in Fig. 10 is also

shown in Tables S1-4 (supplementary material) in order to split the level of variance explained by each climate variable into categories and provide a quantitative perspective. This data shows that there are only 4 regions where half of the models or more show precipitation either fully explains or overexplains the full-climate-driven variation: Svalbard, the Russian Arctic, South Asia West, and South Asia East. There is also only one region in which temperature fully explains or overexplains the full-climate-driven variation in fewer than half of the models: Central Asia. Overall, precipitation only explains a minimal amount

of full-climate-driven variation, with a small number of outliers, while the proportion of variance explained by temperature differs more between models and regions. In Tables S3 and S4, each climate model shows a similar overall distribution across relative variance categories for both precipitation or temperature, so the OGGM response to climate signals is similar across all 6 models.

## 5 Discussion

The normalised regional length results vary considerably between regions and between climate models within each region, but the most consistent trend is a majority of regions demonstrating some form of discernible industrial retreat for a majority of climate models. This is reflected by the Leclercq observations also demonstrating industrial retreat in a majority of regions, with the observations typically showing a relative retreat that is similar to the upper end of modelled retreats. There are 7 regions where the observed retreat is within the range of modelled retreat: Alaska, Western Canada/US, Greenland Periphery, Scandinavia, Central Europe, Caucasus/Middle East, and Central Asia, though in some cases the observed retreat is at certain points slightly steeper than the range of modelled retreats. The results in Central Europe are also consistent with those in Goosse et al. (2018), the forerunner to this study also using OGGM. In North Asia, Low Latitudes, and Southern Andes, we see trends of retreat over the 20th century in all models and in the Leclercq observations, but all of the models underestimate the retreat shown in the observations. Amongst the other regions, there are those where the modelled lengths are just too inconsistent to draw conclusions on where the observations sit within the modelled range (e.g. South Asia West), those where the observations show distinct features which are not present in the modelled trends (e.g. South Asia East), and those where there is neither consistency between models nor between the observations and any modelled lengths (e.g. New Zealand). However, it is difficult to find much consistency between regions where the observations and modelled lengths match poorly as the features appear specific to each region.

While the use of normalised glacier lengths removes the ability to tell which models result in longer or shorter glaciers at the end of the model run, and how these compare to observed lengths, it does allow the differences in responses to climate change trends between models to be seen more clearly. When the model results are highly stratified, such as in Central Europe, Southern Andes, or Caucasus/Middle East, this indicates that the differences between the results from each climate model can be attributed to significant differences in the climate variables for each model. In many cases, the stratification is the result of varying start years and severity of recent retreat. Results using the CESM and GISS models often have lower pre-industrial relative glacier length relative to 1950, indicating smaller and/or later starting industrial glacier retreat, while IPSL and BCC-CSM have high pre-industrial length relative to 1950, and show more pronounced and/or earlier-starting retreat; given the many regions where the observed retreat is at the upper end of the range of modelled retreats or even exceeds this range, IPSL-driven and BCC-CSM-driven lengths are more often a better match for the observations. As these differences are common between a number of regions, it indicates differences between the climate model data on scales greater than that of individual glacierised regions.

Variable start dates for the observed length change timeseries, with a number of regions lacking any pre-industrial representation, make comparisons with model results difficult for the pre-industrial period. Trends can be seen in the pre-industrial model output for a number of regions, but they are smaller in magnitude, and less coherent, both between climate models and between regions, consistent with the lack of global-scale temperature trends prior to industrial warming (Neukom et al., 2019). This is explained largely by the comparison of the default model output and the fixed climate runs (see below). In particular, for the runs using constant temperature (Fig. 9), the divergence between the constant temperature and the full climate run typically

occurs in the industrial period. As the most coherent changes both in climate model variables and in model results occur in the last ∼150 years, we do not examine the patterns of pre-industrial length change on a per-region basis.

In the constant precipitation/climate runs, in almost all cases, temperature is the dominant forcing, explaining much more of the variability in glacier length than precipitation (Fig. 8, 9, 10). The sum of variability explained by temperature and precipitation individually rarely matches the total variability. The common phenomenon of the temperature-only forcing showing greater variability than the full climate runs suggests negative feedbacks between temperature and precipitation have an effect on overall glacier geometry change. This demonstrates the importance of using dedicated glacier models in predicting past

glacier changes, as simple temperature and precipitation proxies cannot properly capture this behaviour. A notable feature of Fig. 10 is the prevalence of relative variance values greater than 1, including some problematic values of 2 and above. If the values of temperature and precipitation were statistically independent in the climate model data, and OGGM's responses to temperature and precipitation were independent, we might not expect relative variance values greater than 1, as OGGM's output for the full climate runs is a response to both temperature and precipitation changes simultaneously. However, there

are several possible reasons for the observed high relative variance values, and different region/climate model combinations indicate different such reasons. Another study which uses a similar mass balance model to examine glacier sensitivity to climate change (Marzeion et al., 2014) finds that predicted future precipitation changes somewhat dampen expected mass losses due to temperature increases, and a similar effect may play a role in cases where we find relative variance greater than 1 for temperature-only forcing. In some cases - such as Central Asia in Fig. 9 - where glaciers take a long time to reach equilibrium

even after the spin-up, differing rates of approach to equilibrium in the full and constant precipitation/temperature runs can cause large differences in variance; this is purely a modelling issue. In others - such as South Asia East and New Zealand in Fig. 8 - the constant precipitation run shows greater overall retreat than the full run, which is the primary cause of greater relative variance. In the South Asia East case, it seems the precipitation in the 1400-1450 climatology used for the constant precipitation run is lower than the average value for the full model run, gradually increasing the gap between the full and

constant precipitation runs; in the New Zealand case, there is an increase in precipitation in the last 200 years which offsets some temperature-based retreat, which does not have an impact in the constant precipitation case. For each region/model combination, the reasons for the relative variance can be different, and it can be difficult to conclusively describe one factor or collection of factors that explains this difference.

Comparing the generally poor correlation between modelled and observed absolute individual 20th century glacier length

changes in Fig. 7 with the reasonable representations of recent retreat in the normalised means of several regions in Fig. 3 & 4, we determine that OGGM - in the configuration used here, and over a millenial timescale - struggles to reproduce the idiosyncrasies of individual glacier evolution in quantitative terms, but does manage to capture the qualitative response to sufficiently large-scale climate trends on regional scales. This is fairly consistent with OGGM's fundamental design; location-specific input data for each glacier is deliberately limited, so a number of processes that are highly localised to individual

glaciers (e.g. shading, snow drifting) cannot be represented, but responses to temperature and solid precipitation that are expected to dominate on regional scales are calibrated to match SMB values interpolated from available measurements. High

agreement on a per-glacier basis is not expected in a 'naive' application of the model where no specific additional calibration for reproducing longer-term length changes is performed after the default calibration of SMB.

## 6 Limitations and Extensions

There are a number of limitations on the modelling process which affect OGGM's output in the runs described in this paper and contextualise the results we are able to obtain. These include issues specific to OGGM, issues which exist in all per-ice-mass glacier modelling, and issues which are the result of data availability.

A significant difficulty for OGGM and other per-glacier models of glacier evolution is the treatment of interactions between ice masses which may be separate and conjoined at different points in the glacier's evolution. This covers glaciers that were historically connected but which have separated due to retreat, (less commonly) glaciers that were historically separate but which have connected due to advance, and glacier complexes which represent dynamically separate ice flows but which are physically connected. It is possible within OGGM to merge glacier flowlines in the setup of an experiment, but when OGGM dynamically models the evolution of a glacier, only the ice mass of that specific glacier is considered, and therefore it is implicitly assumed that there are no other ice masses nearby. Particularly in the case of glaciers which are today small and nearby within the same catchment, and are hindcast for conditions colder than at present, the past glacier states - modelled individually - may overlap, suggesting that in reality they would have been part of the same glacier and dynamically connected. The issue of contiguous ice masses being made up of multiple dynamically distinct flowing glaciers is a matter of data availability as well as of model ability. RGI makes divisions of individual ice masses into separate glaciers where it is deemed appropriate, based on identifying ice divides from DEM data, but this is not based on a sophisticated physical understanding of ice-flow divisions. In OGGM's case, contiguous ice masses with divided flows cannot be modelled simultaneously. Any attempt to address ice mass interactions would considerably increase computational complexity by requiring the simultaneous modelling of all nearby glaciers, even before the demands of simulating interactions. The further from current glacier conditions a modelled glacier is, the more impactful these concerns are; it is therefore a larger issue in this paper's millennium-scale modelling than in the century-long period for which the model is calibrated. However, because in many regions, peak average glacier sizes are not larger in the last 1000 years than in the last 200 years, the risk of greater errors due to historical interactions between currently distinct glaciers does not necessarily increase much when extending the modelling period back beyond the first onset of recent retreat.

Limited glacier observations beyond those available from glacier outlines are ubiquitous in glaciology, and this is compounded for longer historical timescales as available data become even smaller. As Leclercq et al (Leclercq et al., 2014) is likely the best homogenous and quantitative set of long-term length records for glaciers across many regions, and is still sparse both spatially and temporally, we do not feel that we have sufficient data available to both calibrate to length changes and compare against a separate set of length changes. We do, however, see benefits to considering the performance of a 'naive' model setup, whereby we do not tailor the setup of the model to reproduce one particular variable of interest. By calibrating to a certain variable, we can guarantee that the model produces results that are relatively 'correct' on the largest scale even if it is

not the result of significant model skill, which is not the case if we allow the model to work in a way which is agnostic of the variables we will be examining from it.

    For historical glacier states, which are typically larger than modern glaciers, there exists an issue of determining the geometry of the glacier when it extends beyond the modern tongue. In OGGM, the flowline for the initialised (modern day) glacier is based on an algorithm applied to the glacier outline and local DEM, while the estimation of a below-glacier-terminus flowline

- used if the glacier grows beyond its initialised length - comes from a relatively simple iteration on the DEM gradient near the end of the glacier based on the idea of flowing downhill as efficiently as possible. This method can struggle to deal with cases where glacier dynamics may cause the glacier to flow in ways which are not necessarily in the direction of steepest local gradient (e.g. heading over a lip of rock that is in the direction of existing ice flow). Naturally, examples of glaciers with periods of observed advance are relatively rare, and where they exist are often the result of unique processes like surging which

OGGM does not handle well, so the evidence base for proving or disproving the effectiveness of OGGM's below-modern-terminus flowlines is limited. As this paper deals with glaciers in conditions where glaciers are considerably larger than the present, this adds uncertainty compared to modelling for modern or warmer-than-modern conditions.

    The response time of glaciers to changes in climate - and the different response times that glaciers can have at different sizes - impact our ability to precisely compare the timing of glacier responses under different GCM forcings. We know the

response time does introduce an amount of lag to the responsiveness of length changes, but we cannot directly ascribe a single response time value to an individual glacier that is invariate through geometry changes over time and through different types and timescales of climate variation. This means that even when there is a qualitatively similar change in climate across multiple GCMs (and the real climate, where relevant) - say, the onset of a warmer period - glacier responses under one GCM can be faster or slower than another depending on the glacier geometry before the change. Our use of normalised glacier lengths is

intended in part to allow comparisons in responses to changes in climate even where the lengths of glaciers under each GCM forcing differ, but this cannot account for changes in the speed at which these changes happen. Where changes are rapid, or happen in quick succession, this is particularly impactful. Differing response times are part of the reason it may not be valid to ascribe much significance to smaller differences in the inflection points determined in Fig. 5, for example.

    There exists a specific issue with OGGM which causes upward spikes in glacier length in certain situations. This is evident

for several regions in Fig. 3, particularly Svalbard and Central Asia. For very small glaciers, in years with signficantly higher than average accumulation and/or lower than average ablation, OGGM can grow glaciers by considerably extending the tongue of the glacier if net accumulation is positive further down the flowline. In a physical sense, even if a colder year does result in accumulation of snow well in front of the previous glacier front, this is less representative of glacier advance and more a matter of an adjacent, possibly multi-annual snow patch which is not dynamically connected to the glacier itself. OGGM will

typically remove the added glacier length quickly, as the added glacier area is at a lower elevation and therefore has a more negative mass balance, so longer-term trends in glacier length are not affected, but it can raise the variance in glacier length in some cases. Some regions have a signal in the mean modelled glacier length that likely results from this problem, in particular Central Asia and Svalbard.

This paper deals specifically with a subset of glaciers for which historical length measurements are available. This provides
the longest term available dataset against which to compare OGGM's results, but is only a small subset of glaciers globally
(RGI Consortium, 2017). For perspectives which are more comprehensive - albeit with less data to verify against - it will be
necessary to model a full inventory of glaciers in each region. In particular, an attempt to model all glaciers in a region (even
for an incomplete inventory) allows estimates of total ice mass change and corresponding changes in sea level. While the much
larger glacier set is a heavy multiplier on required computing resources, the experiment design could be essentially identical to
that described here. However, the typically much smaller glacier sizes in the RGI as a whole compared to the Leclercq length
data (Fig. 1) do suggest inaccuracies that have a greater impact at smaller glacier sizes (such as the length spiking mentioned
above) will be amplified. Along with the potential underrepresentation of small glaciers even in the RGI (Parkes and Marzeion,
2018), this means the sensitivity of small glaciers will be critical, and should be given special consideration.

## 7   Conclusions

We complete modelling of glaciers in 16 of 19 RGI regions over the 1154 year timescale, with minimal model failures due
to problematic glacier growth (growing uncontrollably or shrinking to nothing) or other reasons. In several regions, OGGM
is able to reproduce substantial qualitative regional average length changes during the period of observational record for the
full set of 6 GCM inputs used, and in some cases one or more of the GCM inputs results in quantitative length changes which
are close to the measured regional changes. In these cases, the observed recent retreat is typically at the top end of the set of
modelled retreats. Regional result comparisons are heavily dependent not only on the modelling skill of OGGM and the quality
of the GCM reproductions of real recent climate, but also on the number of glaciers available to form each regional average
and the noise in both the model outputs and in the observations due to varying numbers of contributors to observed regional
means over time. Through use of a split regression to identify turning points in the regional modelled timeseries, we find that
in many regions the feature which is most obvious and most coherent across different GCM inputs in the trends over the last
millennium is the transition from constant or modestly increasing glacier lengths in the pre-industrial period to steeper recent
glacier retreat.

We do not find that the application of OGGM here produces a good match between modelled and observed retreats on the
scale of individual glaciers. There is no overall bias in the 1950 glacier lengths produced with 4 of the GCM outputs, with small
biases relative to the range of differences in the remaining 2, but for all GCM inputs there is at best a weak correlation between
modelled and observed changes over the course of the 20th century. We therefore suggest that while OGGM can be used to
understand trends on broader scales, it is not reliable for individual glaciers over this timescale without additional, specific
calibration.

Model runs driven by temperature and precipitation individually show that for almost all GCM inputs in almost all regions,
temperature is the primary driver of modelled glacier length variability. However, OGGM's response to climate forcing is a
matter of some complexity, despite the ostensibly straightforward mechanics involved in the way the model calculates glacier
ablation and accumulation. As a result the fully forced runs cannot be understood as a simple function of the temperature-

forced and precipitation-forced runs. The existence of several cases of temperature-only runs showing dramatically higher glacier length variation than fully-forced runs suggests an issue within the modelling process that should be addressed in order to have greater confidence in OGGM's results in the regions where this occurs.

Given the apparent suitability of OGGM for reproducing trends across broader sets of glaciers - despite a lack of confidence in per-glacier results, and with modelling concerns still to be addressed - the next step is to attempt to model entire global glacier inventories over similar timescales, with a particular focus on total volume change as a contributor to sea level change.

*Code and data availability.* Code to run OGGM (Maussion et al., 2019) is available at http://oggm.org along with supporting documentation. Data on matching between Leclercq and RGI glaciers is available as part of the code repository for OGGM, at https://github.com/OGGM/oggm-
470 sample-data/tree/master/leclercq.

*Author contributions.* Study concept devised by HG. Model runs and analysis performed by DP. Manuscript written by DP with contributions by HG.

*Competing interests.* The authors declare that they have no conflicting interests.

*Acknowledgements.* This work was supported by Fonds National de la Recherche Scientifique (F.R.S.-FNRS-Belgium) in the framework of
475 the project "Evaluating simulated centennial climate variability over the past millennium using global glacier modelling" (grant agreement PDR T.0028.18). Hugues Goosse is Research Director within the F.R.S.-FNRS. We acknowledge the World Climate Research Programme's Working Group on Coupled Modelling, which is responsible for CMIP, and we thank the climate modelling groups for producing and making available their model output. For CMIP, the US Department of Energy's Program for Climate Model Diagnosis and Intercomparison provides coordinating support and led the development of software infrastructure in partnership with the Global Organization for Earth System Science
Portals. We are grateful to Fabien Maussion and the rest of the OGGM development team for providing ongoing support and advice on model usage. We thank Ben Marzeion and 3 anonymous referees for reviews that have considerably improved the finished manuscript.

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

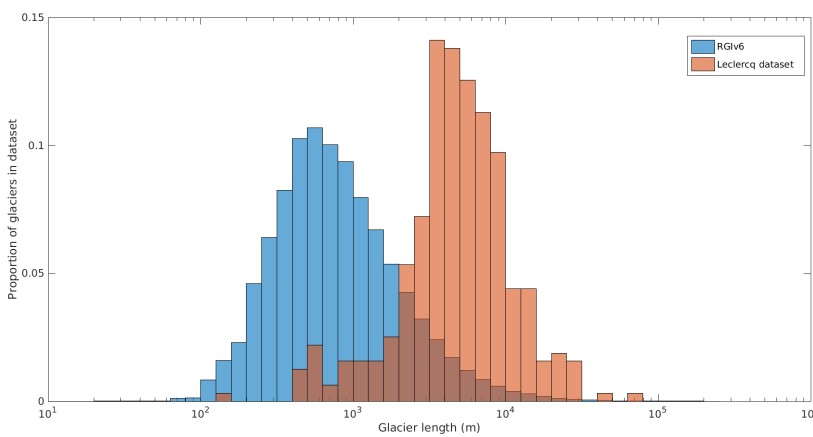

**Figure 1.** Distribution of 1950 Leclercq et al. (2014) dataset glacier lengths (for the 412 glaciers that are matched with RGIv6 glaciers) vs distribution of RGIv6 most up-to-date glacier lengths. Relative frequency is used due to the orders-of-magnitude difference in glacier numbers in each dataset.

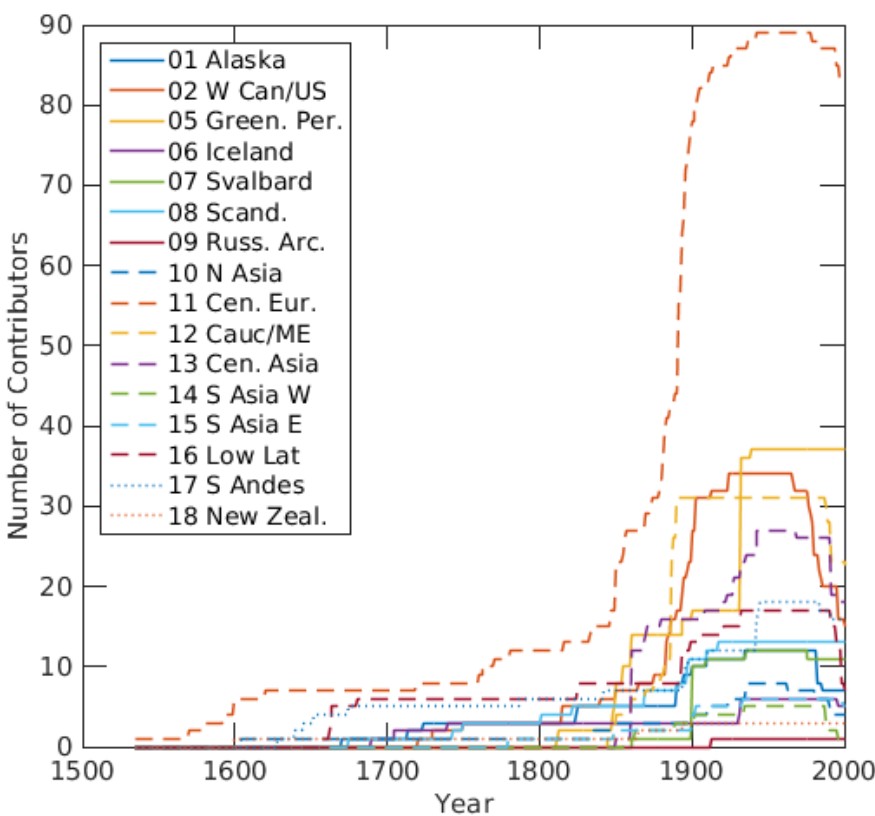

**Figure 2.** Number of glaciers contributing to the Leclercq et al. (2014) dataset regional mean (Fig. 3 black line) over time, by RGI region ID. This does not include the 91 glaciers which are excluded from the regional means for being marine- or lake-terminating, so the total number of glaciers shown here is 321.

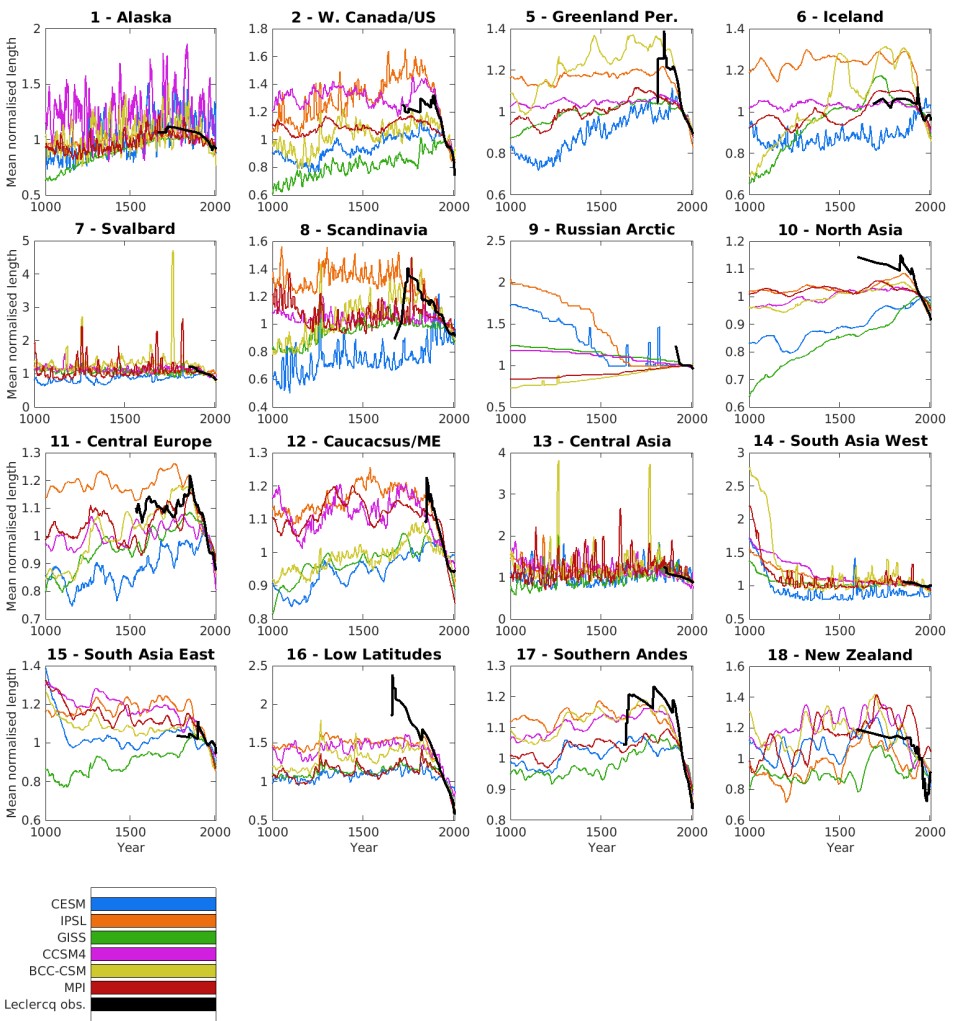

**Figure 3.** Length changes from 1000-2004 CE across 16 RGI regions, modelled by OGGM using 6 separate GCM products (CESM, IPSL, GISS, BCC-CSM, CCSM4, and MPI), and compared to length change observations from the Leclercq et al. (2014) dataset. Each glacier in both the OGGM runs and Leclercq observations has its length changes normalised relative to the 1950 length in the run output or observations respectively. The number of glaciers that contribute towards the mean in each region are listed in Table 1.

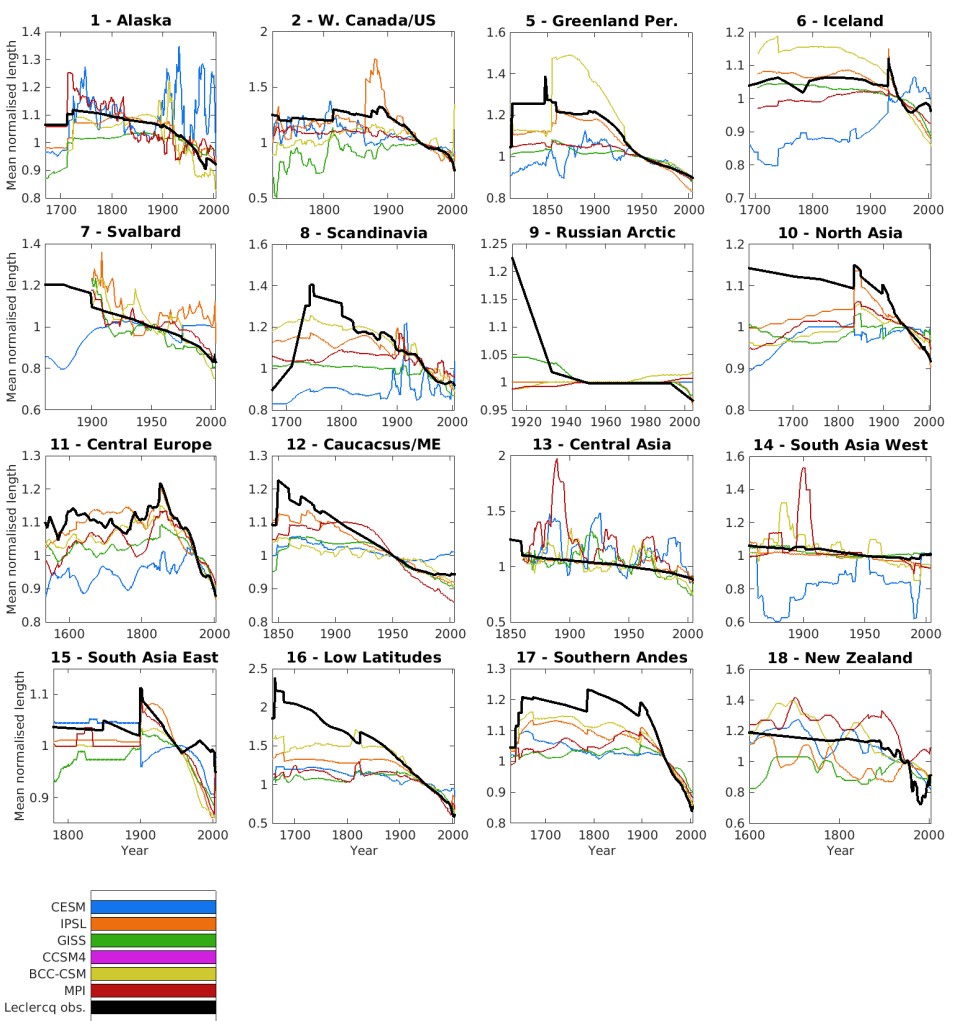

**Figure 4.** Length changes for the period of observational record represented by the Leclercq et al. (2014) glaciers in each region across 16 RGI regions, modelled by OGGM using 6 separate GCM products (CESM, IPSL, GISS, BCC-CSM, CCSM4, and MPI). Unlike in figure 3, the modelled mean in each year includes only the glaciers for which Leclercq data covers that year, resulting in some of data artefacts that come from glaciers entering or leaving the set that contributes to the regional mean being reflected in both the observed and the modelled means.

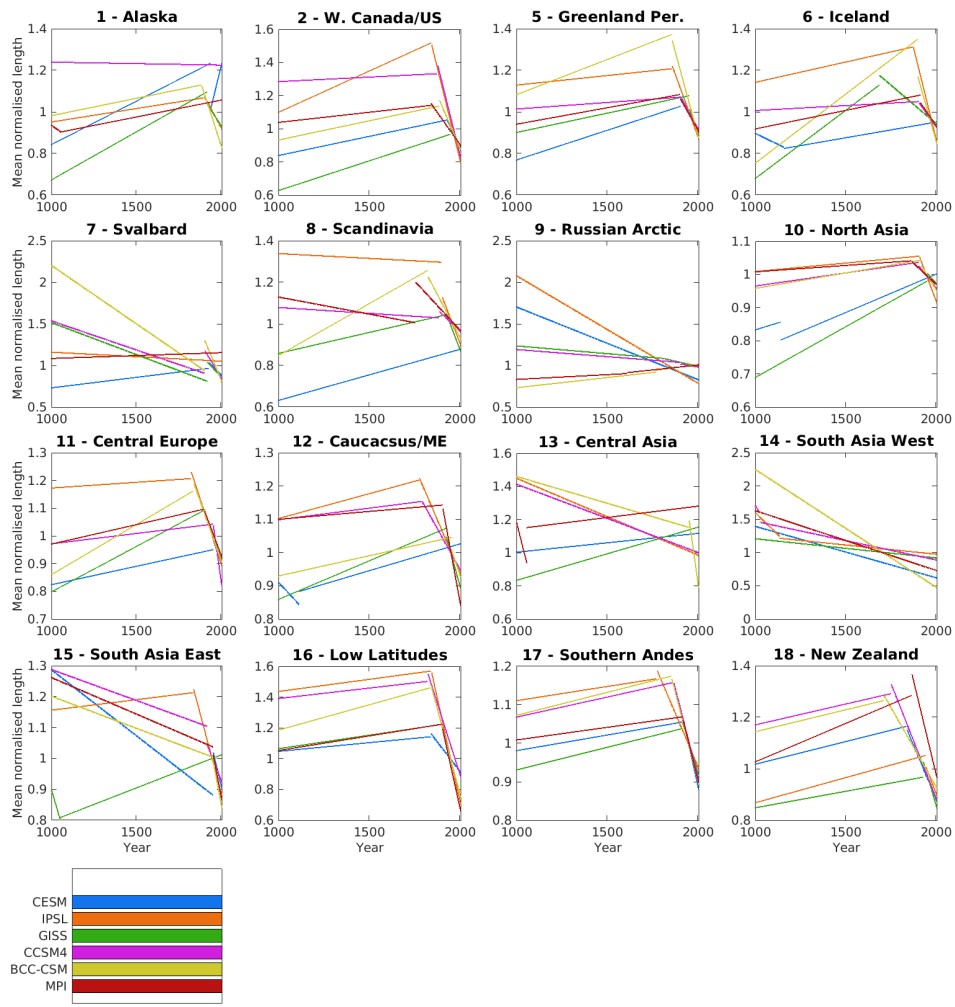

**Figure 5.** Regional length changes represented by an optimised 2-part split regression. For each modelled timeseries of mean normalised glacier length, a split is produced for each year between 901 and 1954 and a simple linear regression performed separately for years up to and including the split year, and years after the split year. The year which maximises the sum of the $r^2$ values for these two regressions is considered optimal, and both regression lines are shown for the optimal year, for each region and climate model.

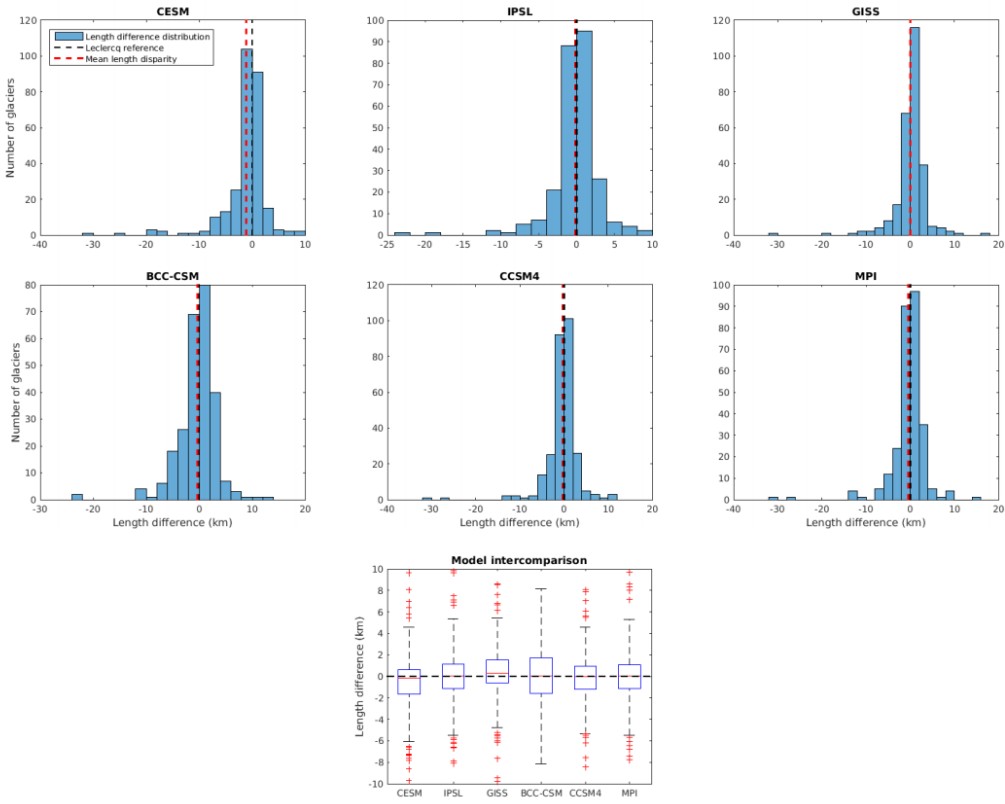

**Figure 6.** Distribution of the errors in absolute modelled glacier length in 1950. Data is displayed per-GCM for all modelled glaciers globally, rather than per-region due to small sample sizes in several regions. The model intercomparison panel shows box plots with the box representing interquartile range.

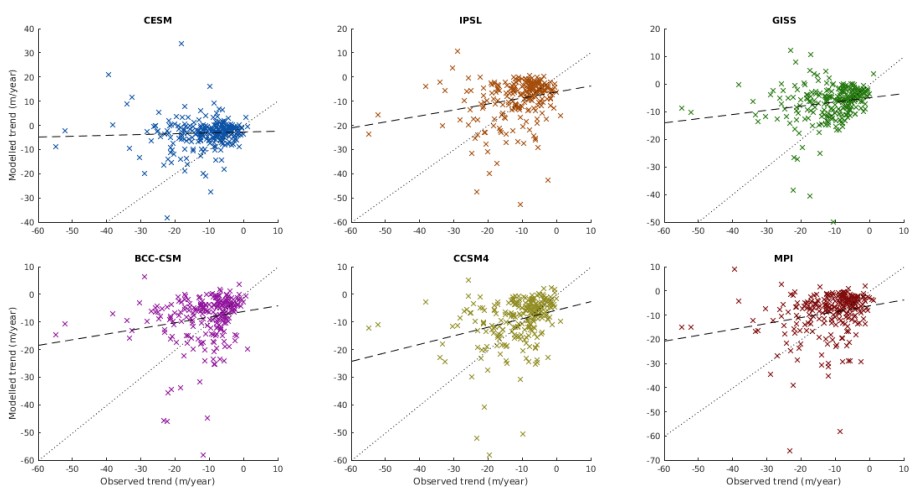

**Figure 7.** 20th century length change trends: per-glacier modelled trends for OGGM runs driven by each GCM dataset vs Leclercq et al. (2014) observations. Dashed black line is line of best fit, and dotted black line shows the 1:1 ratio. Trends are calculated for all glaciers with at least 68 years in the 20th century covered by the observation period (68 years rather than the entire century chosen to ensure more than 90% of glaciers are included). For glaciers with an observation period that starts before 1900, the trend is calculated only from 1900 onward, and for glaciers with an observation period that extends beyond 2000, the trend is calculated only up to 2000.

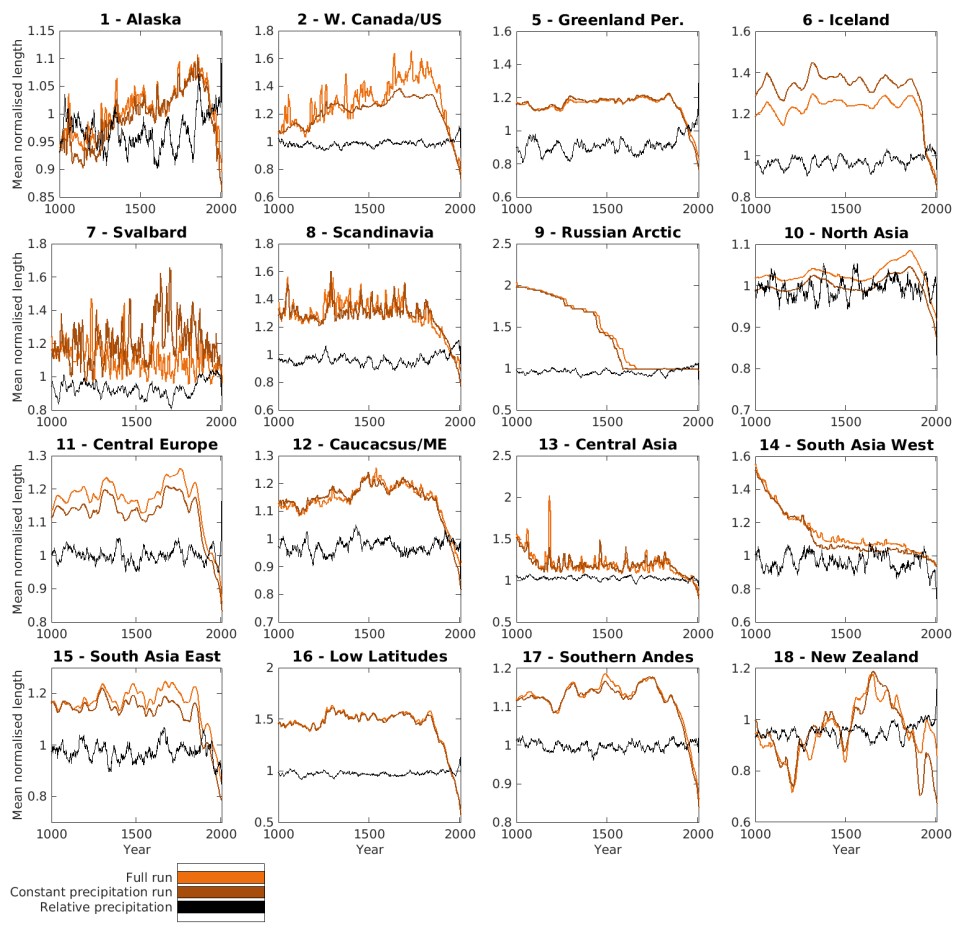

**Figure 8.** Regional length changes for 1 model (IPSL) comparing fully climate run and constant precipitation run, shown with relative precipitation (annual precipitation normalised to 1900-1950 mean climate). The constant precipitation run lengths are normalised to the full climate run 1950 length, to better illustrate differences.

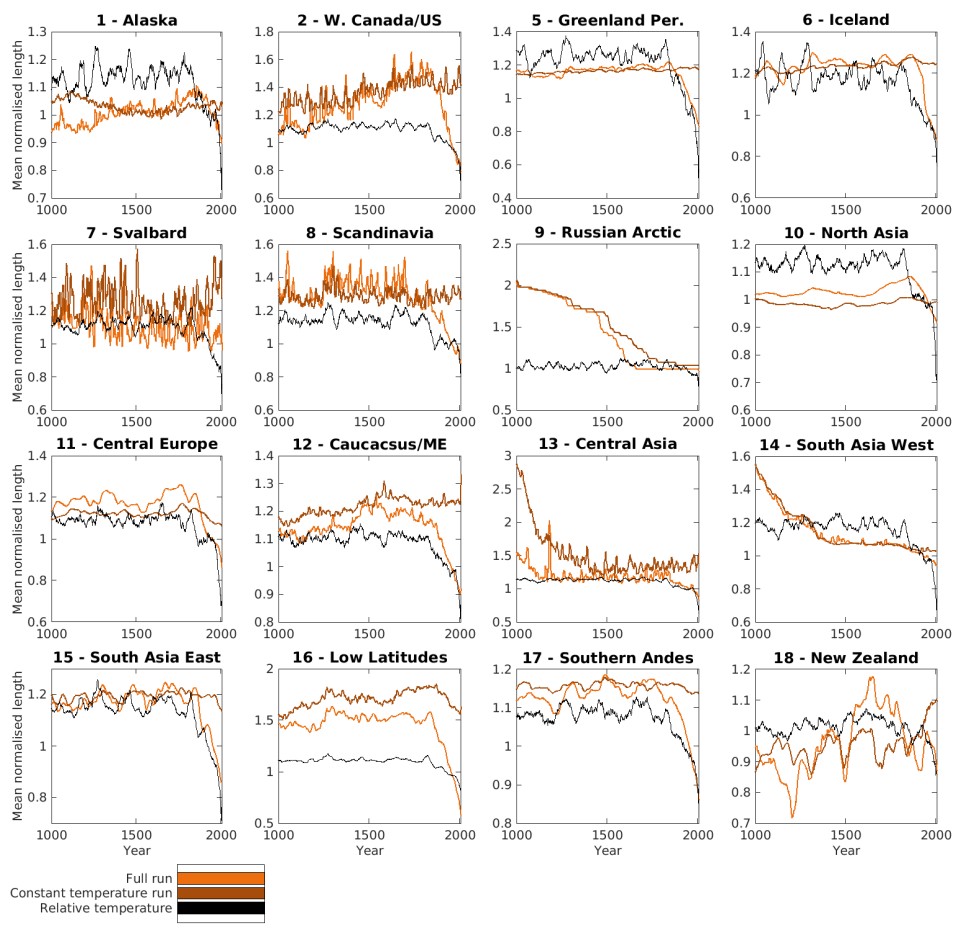

**Figure 9.** Regional length changes for 1 model (IPSL) comparing fully climate run and constant temperature run, shown with relative melt-relevant temperature (annual degree-day sum normalised to 1900-1950 mean climate). Relative temperature is inverted, so that the direction of any trend corresponds to the expected impact on glacier length. The constant temperature run lengths are normalised to the full climate run 1950 length, to better illustrate differences.

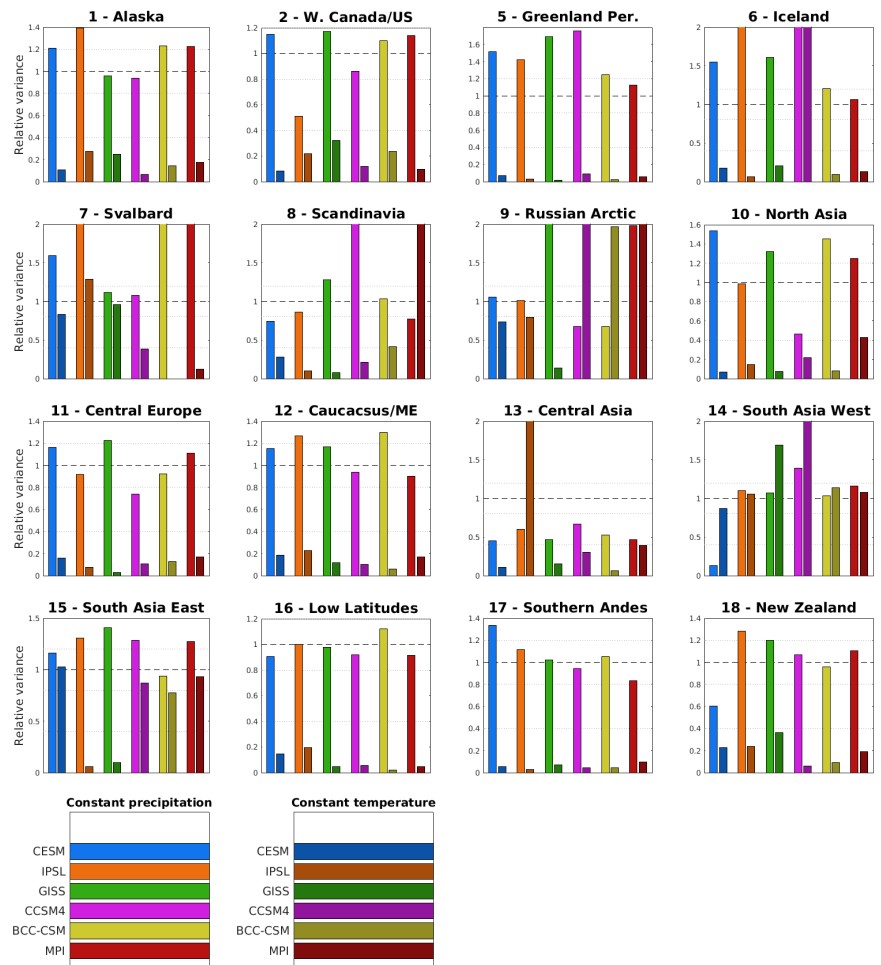

**Figure 10.** Relative variance of the constant precipitation and constant temperature runs to the full climate model run for each of the 6 climate models used. A relative variance of 1 (dashed line) indicates the same overall variance in the constant prec/temp run as in the full climate run. A low relative variance in the constant precipitation run indicates a small impact of temperature on glacier length changes, and vice versa. A high relative variance in the constant precipitation run may indicate a large impact of temperature on glacier length changes (and vice versa), though in cases of relative variance greater that 1, there are several possible explanations for this behaviour. The same information is shown numerically in supplementary Tables S1-4, sorted into categories - the borders of these categories are shown here by the dotted lines.

**Table 1.** Breakdown of the glaciers modelled by region, showing the two reasons glaciers may be removed from contributing to regional means: the glaciers being flagged in the RGI as marine- or lake-terminating, and a failure in the modelling process for all 6 GCMs. For each GCM, the set of glaciers that contribute to the regional mean length changes (any failures for that specific GCM also removed) is shown. Total glaciers in the RGI are shown for comparison, to provide an idea of how well-represented each region is in the Leclercq dataset.

| Region | Glaciers (RGI) | Glaciers (Leclercq) | Marine/lake-terminating | All GCMs fail | Glaciers contributing to mean | | | | | |
|---|---|---|---|---|---|---|---|---|---|---|
| | | | | | CESM | IPSL | GISS | BCC-CSM | CCSM4 | MPI |
| Alaska | 27108 | 20 | 8 | 0 | 12 | 11 | 11 | 11 | 10 | 12 |
| W. Can/US | 18855 | 34 | 0 | 0 | 32 | 30 | 34 | 30 | 34 | 29 |
| Greenland Per. | 19306 | 74 | 37 | 0 | 37 | 28 | 32 | 34 | 31 | 34 |
| Iceland | 568 | 6 | 0 | 0 | 6 | 4 | 5 | 4 | 4 | 5 |
| Svalbard | 1615 | 15 | 3 | 0 | 12 | 11 | 11 | 12 | 10 | 11 |
| Scandinavia | 3417 | 14 | 0 | 1 | 13 | 13 | 13 | 13 | 13 | 13 |
| Rus. Arctic | 1069 | 13 | 12 | 0 | 1 | 1 | 1 | 1 | 1 | 1 |
| North Asia | 5151 | 8 | 0 | 0 | 8 | 8 | 8 | 8 | 8 | 8 |
| Cen. Europe | 3927 | 89 | 0 | 0 | 89 | 82 | 86 | 61 | 86 | 88 |
| Cauc./M.East | 1888 | 31 | 0 | 0 | 31 | 29 | 29 | 30 | 30 | 28 |
| Cen. Asia | 54429 | 27 | 0 | 0 | 27 | 26 | 27 | 26 | 27 | 27 |
| S Asia West | 27988 | 5 | 0 | 0 | 5 | 5 | 5 | 5 | 5 | 5 |
| S Asia East | 13119 | 6 | 0 | 0 | 6 | 6 | 6 | 6 | 6 | 6 |
| Low Latitudes | 2939 | 17 | 0 | 0 | 17 | 10 | 17 | 17 | 17 | 14 |
| S Andes | 15908 | 50 | 31 | 1 | 18 | 18 | 18 | 18 | 17 | 18 |
| New Zealand | 3537 | 3 | 0 | 0 | 3 | 3 | 3 | 3 | 3 | 3 |
| Global | 215547 | 412 | 91 | 2 | 317 | 285 | 306 | 279 | 302 | 302 |