# Peer review of "Modelling regional glacier length changes over the last millennium using the Open Global Glacier Model"

_The Cryosphere, 2019_

## Referee Comment (RC1) · Anonymous Referee #1 · 8 Jan 2020

Summary and comments on the manuscript entitled

**Modelling regional glacier length changes over the last millennium using the Open Global Glacier Model**

presented on 15.11.2019

by

D. Parkes and H. Goosse

**Summary**

In this manuscript, the authors shed light on glacier evolution in the past millennium using the Open Global Glacier Model (OGGM v1.1; Maussion et al. (2019). The authors focus on 339 land-terminating glaciers with multi-centennial length records covering most glacierised regions in the world. The ensemble is driven by temperature and precipitation from six general circulation models (GCM). Prior to the forward simulations, glacier units in the length-record data set are matched with the entries in the Randolph Glacier Inventory (RGI). Comparing the millennial simulations with the length records, the authors conclude that modelled length changes are consistent with the observations. Keeping either the temperature or the precipitation forcing fixed, the simulations show that temperature explains the larger share of the variance in the glacier-length observations.

I was very excited about this study because the authors evoke that they will shed light on the consequences of either calibrating glacier system models during the last century of general retreat or during multiple centuries showing phases both of advance and retreat. Unfortunately, the manuscripts leaves this interesting question unanswered. It seems symptomatic that, although the authors raise high expectations, they are not able to draw strong conclusions. I was all the more disappointed about the technical implementation of the experiments. Important details of the climatic forcing are unclear, the model calibration appears insufficient and the computation of relative regional glacier length changes seems inappropriate. All this precludes a meaningful interpretation of the results. Therefore, I cannot recommend this article for publications in *The Cryosphere*. I fear that the necessary revision of the manuscript will involve fundamental changes in the experimental details, in the analysis and the manuscript content that will justify a new submission. Nonetheless, I leave it to the discretion of the editor if he wants to continue to consider this submission for publication.

**General comments**

**Climatic forcing & calibration**
The authors explain that they use 6 general circulation models (GCM) to drive

OGGM with temperature and precipitation. I suppose that they used monthly fields of surface air temperature and precipitation. Yet no details are given. Is the OGGM-inherent interpolation scheme used for these variables. I think that it is worth to specify the assumed lapse rates. However, my primary concern is the casual description of the climatic forcing. In lack of details, I assume that, after the model calibration, the GCM forcing is directly given to OGGM. Yet the GCM performance will differ around the globe making them more or less suitable for explaining length changes in various regions. In my view, the general practice is to define a common reference period with a climatologically meaningful length and apply the climatic forcing in anomaly mode. In this way, GCM biases in the recent period can be accommodated. This aspect is even more relevant in terms of the OGGM calibration. As the authors only refer to the original OGGM publication (Maussion et al., 2019), I assume that the temperature sensitivity parameter ($\mu^*$) is automatically calibrated based on the interpolated CRU data set. Correct me if I am wrong. Please specify the calibration time period in OGGM. The automated OGGM calibration implies that when you change to the GCM forcing, the model is not expected to perform well in the recent past, reverting the benefit from the *CRU* calibration. Changing to anomaly modes will help. The other option is to calibrate $\mu^*$ to each GCM. As it stands now, I fear that it is almost impossible to interpret the results.

**Representativeness**
Please specify how representative the glacier sample that has length observations is for each RGI region. An idea could be to present numbers of hypsometric and glacier-area distribution of the glacier sample as compared to the entire region.

**Regional length changes**
You compute the regional relative glacier length changes as the mean of normalised length variations of all glaciers with length observations per RGI region. First, this normalisation overrates the importance of small glaciers. You can see this for Alaska in Fig.1: at several points in time, regional relative length changes exceed 0.5 in less than 10 years which should not reflect the response of the large glaciers. Second, the formation of a mean value is highly susceptible to outliers. Outliers in terms of normalised length changes are expected from the small glaciers in the region because large glacier systems will only show moderate relative length fluctuations. Is it possible to use a more robust measure. I do not think that the median will help, let alone that it will be more informative. Yet a weighting by glacier area might help. In addition, you compute the regional mean on an annual basis only considering glaciers with measurements in that year. In many regions you see abrupt step changes in this regional value, which are likely a relict of this strategy. It is therefore very difficult to interpret the regional length change record because they can either arise from the model response or the changing number of considered glaciers from year to year. It would be highly informative to include a plot with the regional length records in addition to the sample size variations through

time. An idea for removing the 'spike' behaviour is to assume a linear length change between sparse length observations. In other words, would it make sense to linearly interpolate the observed length record to a yearly timeline for glaciers with irregular sampling of length information. Admittedly, my suggestion is not ideal but it removes the sample-size dependence from the regional length values.

**Glacier complexes**

In the past, many of the nowadays separated glacier units in the RGI were part of large complexes comprising several glacier branches. As OGGM treats each RGI unit independently, larger glacier complexes in the past are not allowed to form. How important is this fact for the glaciers you are focussing on. The influence of tributaries might well have been important for past length changes even during the period with length observations. Is it possible to consider this effect in OGGM? If not please discuss.

**Calibration strategy**

Recently, Eis et al. (2019) presented an alternative for the standard initialisation technique in OGGM. They show that it is important to not only consider present glacier length in the calibration but also the full geometry. In this way, the uncertainties in hindcast simulations can substantially be reduced. As I understand it, you use the standard OGGM spin-up which is not intended to reproduce length changes even in the last century. If there is no length change calibration in this period, why would you expect reliable performance over an entire millennium? Please justify. From my perspective, it is a prerequisite that the approach is calibrated against the length change record (Leclercq et al., 2014) to guarantee a certain reliability over multiple centuries.

**Manuscript structure**

The 'Methods & Data' section and the 'Results' sections appear as single entities and they lack some structure. For the 'Methods' section, you could present OGGM with some more details on the inherent calibration. For the 'Data' part, you can specify the RGI and the length record with the pre-processing details. Moreover, the climatic forcing could be described in detail. The 'Results' section is a *mélange* with a discussion. I would mention that in the section title. Also try to introduce some sub-divisions.

**Objectives**

As already mentioned above, you raise high expectations in your introduction but the conclusions appear rather vague (e.g. abstract, L35, L44, etc.). Therefore, I suggest that you better streamline the manuscript on the conclusions that you are able to draw.

**Open Data set**

From your description, I appreciate the effort to link the RGI to the length change

record from Leclercq et al. (2014). I therefore suggest that you provide a look-up table between the RGI-ID and the ID numbers of the length record, specifying 'positive' and 'best-effort' matches as well as not retrievable entries.

**Detailed comments**

**L35** To what challenges do you refer here? You emphasise this point already in the introduction but you leave it vague here. Please substantiate.

**L107-108** From this sentence, I would have expected that Table 1 comprises number of glacier units and glacierised area per RGI-region. Please add.

**References**

Eis, J., Maussion, F., and Marzeion, B.: Initialization of a global glacier model based on present-day glacier geometry and past climate information: an ensemble approach, The Cryosphere, 13, 3317–3335, doi:10.5194/tc-13-3317-2019, 2019.

Leclercq, P., Oerlemans, J., Basagic, H., Bushueva, I., Cook, A., and Le Bris, R.: A data set of worldwide glacier length fluctuations, The Cryosphere, 8, 659–672, doi:10.5194/tc-8-659-2014, 2014.

Maussion, F., Butenko, A., Champollion, N., Dusch, M., Eis, J., Fourteau, K., Gregor, P., Jarosch, A., Landmann, J., Oesterle, F., Recinos, B., Rothenpieler, T., Vlug, A., Wild, C., and Marzeion, B.: The Open Global Glacier Model (OGGM) v1.1, Geoscientific Model Development, 12, 909?931, doi: 10.5194/gmd-12-909-2019, 2019.

---

## Referee Comment (RC2) · Ben Marzeion (Referee) · 13 Jan 2020

Parkes and Goosse present millennial-scale simulations of glaciers to investigate the ability of climate and glacier models to represent long-term glacier behavior including advancing and stable conditions, to evaluate model performance compared to observations, and to study the relative importance of temperature and precipitation anomalies for explaining glacier variability on such time scales.

The study is clearly of interest for the readership of The Cryosphere and definitely has the potential to progress understanding of glacier-climate relations on long time scales, as well as the utility of models to investigate these relations. However, the

full potential of the study is not realized, and with an extension of the analysis (in particular, adding more quantitative measures of model performance) much more could be gained. The objectives are stated, but the abstract and introduction need to be refocused (and potentially adjusted significantly, depending on the authors' choice if and how to implement the ideas I sketch below). Finally, a bit more background is needed on the use of the glacier model.

All in all, I think the paper needs major revisions (detailed below) before it can potentially be accepted for publication in The Cryosphere.

General/major comments:

- I find the focus of the paper a bit unclear. From the abstract, I get two objectives, which are obviously linked, but not spelled out very clearly: (i) test whether calibration of the model during the retreat phase is good enough and leads to adequate results also in times of advance or stability, and (ii) identify whether precipitation or temperature anomalies are more responsible for glacier length changes at multi-centennial time scales. I think it might help to use (ii) as the main objective, which would require (i) as an intermediate step. This also relates to L38-39: "it is important that these models are examined over time periods where more stable glacier geometries were expected". It could be argued that if a model is not foreseen to be applied in conditions where glaciers are stable or advance, the model does not need to be able to show such behavior (e.g., to my knowledge, the representation of ice geometry change of the model of Huss and Hock, 2015, does not provide for advancing glaciers). It would be good to give some explicit reasons why it is important. In the introduction, the attribution of glacier length change to either precipitation or temperature anomalies seems like an afterthought. I think it would help the paper a lot to restructure with one clear objective (which I think could be this attribution – but the authors may disagree).

- It probably would be helpful if a bit more is said on the setup of climate forcing in OGGM: the authors are referring to "level 3" preprocessing of OGGM, but don't give

any details on how OGGM treats climate model output before application to the mass balance model (this concerns bias correction/anomaly coupling, estimation of solid precipitation, any corrections etc). As it stands now, readers might by surprised by the apparent ability of CMIP5-type models to represent mountain climate conditions accurately enough, which is only half the truth. (see also L166, "as provided", which is not true)

- Regional averages are presented and discussed, and this comes at the relatively high cost of having to find a way how to calculate regional length changes (see discussion around L120-125). It is my impression that the calculation of regional averages is a purely graphical requirement, needed in order to avoid having to inspect 339 individual glacier time series. This points to a more significant problem, which is that the assessment of the model results depends too much on this graphic representation. I would recommend the authors to expand the analysis of results to a more quantitative evaluation, such that the assessment of regional differences depends less on visual inspection of graphs that necessarily are associated with shortcomings (such as the spikes resulting from changes in the observational ensemble). At least, it might be worth to extract data from the modeled glaciers at the same time as observations exist (adding a third version to Fig. 1 and 2), so that the modeled regional average would have the same spikes as observations if it was perfect. But I think this would not be the optimal solution. The analysis based on linear trends aims in the right direction, but doesn't really relate to the observations, so it is not helping with this issue.

- The discussion of Fig. 4-6 would form a nice basis to infer something about the adequacy of the climate models if the comparison of the model results to observations was more quantitative. E.g., it would be possible to quantify how much the glacier model performance is reduced (presumably) if either temperature or precipitation information is withheld, giving further insight into their relevance (and the climate model's ability to represent precipitation and temperature evolution – after all, it is possible that a model's performance increases when one of the two variables is held constant). I also think that

the discussion of variances misses the opportunity to say more about physical, climate related causes of regional differences. This is also true for the discussion of relative variances > 1, which basically imply that there is dependency between temperature and precipitation. This is discussed on a technical level (it might be added how the way the solid fraction of precipitation is calculated in OGGM adds to this dependency), but there are also climatological reasons that should be considered here. This should include a discussion of the relevant literature on precipitation vs. temperature influence of glaciers (which is currently almost completely missing). Also, a discussion of Marzeion et al. (2014, DOI: 10.5194/tc-8-59-2014), which includes similar experiments, may be helpful.

Specific/minor comments/suggestions

- L1-2: "observational record for glacier changes falls within the post-industrial period, associated with global glacier retreat": I would suggest to delete "post-", and to replace "associated" by "coinciding"

- L3 (and throughout manuscript): delete "post-" from post-industrial (unless this is a standard term – but to me it sounds like "after the industrial age", which is not what you mean)

- L13-16: suggest break in two (or more) sentences

- L23: rephrase to avoid nested parentheses

- L26: avoid nested parentheses

- L29: avoid citation of webpage (OGGM e.V.) if there is a proper publication (Maussion et al., 2019)

- L32: OGGM also include precipitation in the calibration

- L35: please spell out some of the additional challenges

- L65: suggest to rephrase to "variability over time output of OGGM driven by"

- L78: fix double citation

- L80-81: avoid nested parentheses

- L101: fix citation

- L115-117: "smaller in relative magnitude" yes, but for many applications, the relative magnitude of changes is not that important, but the absolute mass change. So this is maybe not such a big problem.

- L119-120: it is unclear to me here why a mean regional glacier length estimate is needed? Up to here it seems all comparisons to observations are done on a per-glacier-basis (which seems like a better idea to me).

- L 151: from my own experience I appreciate the difficulty of doing the matching, but 38 "not found" glaciers strikes me as a high number. Typically, glaciers with length observations tend to be more "famous" than the average glacier. Might it be worth checking in other data bases (e.g., GLIMS) based on glacier name?

- L 153: please provide a table linking RGI-ID to Leclerq's database as a supplement (as a service for similar, future studies)

- Fig. 1 & 2: I'm surprised that also the model results are very spiky in some regions. Why is that so? It would also be helpful to use the same vertical axis range in all subplots (even if it cuts off parts of some graphs), since the normalization make the regional comparison easier (and the different axis limits make it harder).

- L182-186/Fig 1&2: I don't really understand what makes regions 14, 15 and 18 stand out from the other regions? Again, I think a more quantitative comparison of the model ensemble to the observation ensemble would be very helpful.

- L189: not sure what is meant by "stratified", so I also don't understand the following argument

- L199 and following: would it make sense to do this analysis first for each glacier in

the region, and then build the regional mean? This could result in an indication how robust the estimation of the "inflection" year is, and also give some more information on intra-regional variability of the glaciers' behavior.

- L213-125: I find this argument a bit weak, given that the "inflection" year is probably very sensitive to short-term interannual variability in the climate time series

- Fig. 5: was the non-zero melting threshold temperature of OGGM taken into account when calculating the temperature time series shown here (also, in L226, it says "degree-days" – is it really days, or months)?

- L262-263: it is not only OGGM, but (probably at least as much) the GCMs that are responsible for the level of agreement.

- L270: add citation to Goosse et al. (2018) here

---

## Referee Comment (RC3) · Anonymous Referee #3 · 18 Jan 2020

**Review of 'Modelling regional glacier length changes over the last millennium using the Open Global Glacier Model' by David Parkes and Hugues Goosse**

Submitted to 'The Cryosphere Discussion' on November 15 2019

In this manuscript, Parkes and Goosse, use the Open Global Glacier Model (OGGM) to model the temporal evolution of a few hundred glaciers over the past Millennium. For this, they select glaciers from different regions in the world for which glacier length records are known from observations. The simulated glacier lengths under six different climate forcings are compared to 'observed' length records and the authors suggest that a good general agreement is obtained.

The idea put forward by Parkes and Goosse in this study is an interesting and a challenging one. As they rightly state, many of today's glacier evolution models focus on relatively limited time periods for model calibration and/or evaluation (typically present-day period or recent past) that are shorter than the time periods over which these glaciers adapt to climatic conditions (their response time). This raises questions about their applicability over longer time scales. A study on the multi-centennial to millennial evolution of glaciers and its ability to reproduce observed changes is thus of high relevance and directly relevant for the readership of 'The Cryosphere'. At this point, I do however feel that many of the interesting questions that are raised in the introduction are not really elaborately answered, and that this study is in need of additional quantifications and additional experiments before it can really be considered to be of interest for the glacier modelling community. I made a list of suggestions on how this could be tackled and other issues (some rather major, other minor) that should be considered before this manuscript could be considered for publication.

**General comments**

1. At this point, many statements a bit vague and the analysis presented are not very in-depth. Additional information (mainly quantitative information) and analyses will be required to really make point that the authors are able to use OGGM to closely reproduce past glacier changes. Two main points here:
   a. You chose to mainly focus on relative length changes at regional scales and explain why you present the results in such a fashion. This is definitely fine, but I do not think this should impede you from also giving some results in absolute values and for individually modelled glaciers. This is just a matter of showing the results differently and does not require any additional runs/simulations. More specifically, it would be interesting to see how the model is able to reproduce the present-day glaciers: i.e. are the glaciers that you obtain at the end of your simulation (around 2000) close to the observed ones? So far, you argue that it is important to look at the model for periods when glaciers were more stable (e.g. l.38-40), but this should not stop you from also considering the model performance for the recent past and its capability to closely reproduce the present-day glacier. An example: take a glacier for which the relative length change is well reproduced over the last 300 years, but where your modelled present-day length is 8 km vs. an observed length of 20 km… This means that your glacier was also 2.5 times too long 300 years ago (everywhere between 300 years ago and now): would you argue that the model does a good job at representing the changes here? Would be good to have a figure (e.g. in suppl. Mat.) with on the x-axis the observed glacier length at present-day, on the y-axis the modelled present-day glacier length (after transient run) and having every individual glacier plotted in this (for all regions together; could do this for one climate model)
   b. Role of the SMB. You barely mention the SMB component of the model, which I found surprising, given that this is the main driver for the glacier behavior (the dynamics then translate your SMB forcing – with a lag due to the response time – to a length change). For instance, when considering the role of temperature vs. precipitation forcing, it would be highly relevant to describe how much these components affect the modelled SMB (with quantifiable information). Many studies have provided insights in the role of temperature vs. precipitation forcing for the SMB (e.g. Lefauconnier et al., 1999; Braithwaite & Zhang, 2000; Oerlemans & Reichert, 2000; Sicart et al., 2008; Trachsel & Nesje, 2015), and often found that the temperature is the main driver. Your finding that temperature is the main driver directly results from the calculated SMB, which is far more sensitive to temperature changes than it is to precipitation changes. Would be really nice if you could show some of the calculated SMBs and perform some basic sensitivity tests (e.g. what happens with SMB when forced with +1°C, -1°C, +20% precipitation,…etc.)
2. Title is a bit misleading: when reading 'regional' glacier length changes, I would expect that an entire region is considered. However, glaciers from various regions are selected, which in every case represent only a very small subset of all glaciers in this region. Suggest reformulating this, which could be done

by simply omitting 'regional'. Or should rather mention something like: 'in various regions around the world'

3. Almost all the 'action' occurs in the pre-frontal glacier region (compared to the present-day ice cover): how well is OGGM able to handle this? More information is needed about how the flowlines are defined here, how the cross shape is parameterized,…etc. This information is lacking at the moment.

4. I generally found the manuscript relatively easy to follow and found the figures to be simple, but clear, which is very nice. At several occasions I did however get lost in long sentences (often multiple brackets are being used…) and had to read through these several times before getting the meaning of the sentence. I therefore suggest reducing the use of brackets, and splitting up long sentences where possible. Examples are provided in the 'specific comments' section below.

5. You describe this study as being a kind of first attempt to reproduce past length changes with a flowline model for glaciers in many different regions and suggest that this would open the door to regional scale applications. I agree with the former, but have some doubts about the possibility to fully extend this to regional scales. What about glaciers that are now separate ice bodies but used to be connected? What about glaciers that disappeared by now but may have existed before (whether as separate ice bodies or tributaries to present-day glaciers) – a field in which Parkes himself authored an important study (Parkes & Marzeion, 2018). I think it would be fair to also mention these issues/challenges in your conclusion, were you provide an outlook (last sentence of the manuscript).

**Specific comments**

Abstract
- l.5: 'in active development'. Well, I guess this can be said about almost every model. OGGM has now reached a certain maturity and will of course further evolve in the future, but think it would be better to drop the 'in active development' part.
- l.8-9: 'modelled glacier changes…more rapid than – modelled retreats': quite vague. Try to be more specific (also in previous sentence).
- l.13-16: statement is again rather qualitative here: could you provide concrete numbers that support this statement?

Introduction:
- l.20: Reference to IPCC AR5. For the 'future glacier part', I suggest adding a reference to recent GlacierMIP effort by Hock et al. (2019).
- l.21: most relevant study from Oerlemans to support this statement is Oerlemans (2005)
- l.22: 'direct observations of glacier geometry': what do you consider being a direct observation?
- l.22-26: very long sentence and difficult to follow: suggest splitting up and omitting some brackets were possible.
- l.25-26: 'though even this is likely a significant underestimate (Parkes and Marzeion, 2018)'. Well, the number of glaciers is simply subjective, as it is related to the threshold that is used to decide whether a glacier is mapped (outlined) in the Randolph Glacier Inventory (RGI Consortium, 2017). We know that the number would be higher if smaller ice bodies would also be considered, so would not refer to this as an 'underestimation' here.
- l.29: '(OGGM) (OGGM e.V., 2019; Maussion et al., 2019)': try to avoid multiple repetitions of brackets. Also a few sentences before, in the sentence with 'WGMS' and 'NSIDC'.
- l.32: 'by default calibrates the glacier sensitivity to local temperatures based on CRU data…': what criterion is used for calibration? You mention that CRU data is used, but what do you (try to) match in the calibration procedure? i.e. what is the target? (e.g. measured SMB, geodetic mass balance,…etc.)
- l. 34: 'already experiencing significant retreat… (Zemp et al., 2015)': here I did intuitively expect a reference to work by Leclercq et al. (2014). Becomes clear later in the story that this is the main dataset you'll be using, but nevertheless good to already mention this important study here.
- l.39: 'we expect that…': strange formulation. You expect smaller and globally less consistent temperature trends? Based on what? Or is this just what the reconstructions suggest? Would rather formulate in lines of 'Studies/Observations suggest that temperature trends were smaller and globally less consistent…'
- l.45: when mentioning the glacier length changes, could make link with observed changes from Leclercq et al. (2014) (which you use later, but reader does not know at this point) and Solomina et al. (2016) in which the literature on glacier geometry changes over the last 2000 years is summarized.
- l.45: 'we cannot compare': you as authors? Or the literature in general?

- l.46-47: focus is on the transition from more stable pre-LIA to retreat. This is a complex matter – and many studies have tried to shed a light on this and came up with several possible mechanisms to explain the timing of this transition (e.g. Painter et al., 2013; Lüthi, 2014; Sigl et al., 2018). Would be surprising that your relatively simple setup (with temperature and precipitation forcing only) is able to simulate the right timing (as it is generally known that retreat starts before a real increase in temperatures is observed). Would be good if you could provide a few words of explanation on this.
- l.48-50: observations → any reference for this?
- l.50-52: agree, European glaciers are indeed not representative for worldwide glacier fluctuations. This is clear from recent Glacier Model Intercomparison Project (GlacierMIP), in which a strong contrasting behavior between the evolution of glaciers in various regions is highlighted (Hock et al., 2019).
- l.57-61: long sentence with many brackets, consider reformulating. Furthermore, given the fact that you focus on proxies for past glacier extent, would be relevant to again refer to work by Solomina et al. (2016) here.
- l.68: '…comparisons of between models and differences between regions': which models are you referring to here? Glacier and/or climate models?

Methods and data:
- l.74-75: strange description for OGGM: model for glacier dynamics that accounts for geometry and ice dynamics. Ok, but is also really a model in which SMB is coupled to glacier dynamics to simulate the temporal evolution of glaciers. Would suggest already mentioning the SMB here, and giving more information about the SMB in general in the following sentences, as this is the main driver for your changes over the past centuries... (see general comment 1b)
- l.81-84: you use an uncalibrated version of the OGGM model. What are the implications for the modelled glacier geometries at present-day (i.e. after several centuries of transient run): do you end up having a realistic glacier shape? Would be surprising that this can be obtained without any calibration and by just taking the model as is: see also general comment 1a: would be good if you could show the modelled present-day geometry (after transient run) vs. observed (and thus not only rely on relative changes).
- l.85-89: long sentence: maybe split up?
- Would need information about pre-frontal area and how you treat glacier changes here. See general comment 3 for more information
- l.112: 'may also still be', suggest replacing by 'are': see comment on l.25-26 for explanation.
- l.112-117: in your explanation, you link the glacier response time to its size. Is however not the case for many cases / regions (Raper & Braithwaite, 2009; Oerlemans, 2012; Zekollari et al., 2020), and quite often the main driving mechanism for the glacier response time is the surface slope. Could simply reformulate this by saying that the glaciers you consider are typically large and relatively gently sloping glaciers and that these may not be representative for all glaciers in the region when it comes to their response time, as this is driven by a combination of glacier-specific factors.
- l.118-132: OK to have regional values and relative changes, but also need to show your results in absolute values and for individual glaciers. Does not require additional simulations, just a different and elucidating way of looking at results. See comment 1a for more info.
- l.139: the area given in the Leclercq dataset: area at which time period? Guess this depends on the region/glacier considered? Would be good if could give indication.
- l.140-144: very long sentence. Suggest formulating the part between brackets as a separate sentence.
- l.144-152: very nice to have such a detailed description. Often missing in papers, here very clear. Could potentially even be a bit more specific?
- l.146: 'time of the Leclercq measurement': when is this?
- l.158: Recinos et al., 2018 (= discussion paper) → Recinos et al., 2019 (= final paper)
- l.162: 'excluded per region': was not entirely clear to me. Suggest omitting 'excluded'.

Results:
- l.166-168: you explain that sometimes not fully equilibrated after 300-yr spin-up: if this is the case, why do you simply not consider a longer spinup (> glacier response time) of e.g. 1000 years? Seems that in any case an initial adjustment will occur, because glaciers are never entirely in steady state, and you use this as a starting point. This is OK, but could reformulate this.
- l.180: 'models underestimate the retreat shown in the observations': how come? Maybe also role for other factors not accounted for in your model: e.g. role of aerosols (global dimming) and other mechanisms?
- l.187: '…use of normalised glacier lengths removes the ability to tell which…': well, you can simply also additionally give your results as non-normalised glacier lengths. Need to do this to increase insights

in your results and capability of OGGM to reproduce the present-day glaciers (after multi-century transient run): see general comment 1a.

- l.213-215: 'This suggests that differences in total post-industrial retreat are more influenced by differences in when the retreat starts than by…': OK, from the modelling perspective. Is this also the case when considering the observations?

- l.223: '…we do not examine the patterns of pre-industrial length change on a per-region basis': why not? Would be interesting to examine this and this aspect would add novelty to the paper. In the end, the manuscript almost solely focuses on post-industrial time period (although not clear if modelled absolute glacier length changes are realistic, see comment 1a) and when the retreat starts here, although the simulations cover a millennial timescale..

- l.226: Results shown only for 1 climate model: definitely ok, but would be good if you could argue / give a reason why IPSL is chosen (without this explanation this seems rather arbitrary).

- l.224-256: in general a lot of rather qualitative statements are made at the end of your result section, which does not leave the reader with real take-away messages: i.e. what should one remember when reading this section? Two suggestions for topics to focus on / include in your discussion:
  - I think it is important to focus more on the SMB here and link this to other studies in which the SMB and its sensitivity to temperature and precipitation changes are described + show this for OGGM's SMB component used in this study (see general comment 1b + comment on l.74-75).
  - What about the response time? This is not really mentioned in your story, but in the end this plays an important role, as the response time will explain the lag between the change in SMB (the effect signal of your climate input) and the change in glacier geometry (the glacier length you consider for model evaluation). Would put more emphasize on response time in this section and e.g. explain inter-region and inter-glacier differences in the timing of the post-industrial retreat and how this can be linked to response time.

Conclusions:
- l.261: 'this observed retreat is within the range of the modelled retreats': difficult to judge as only relative changes are given and over regional scales. See comment 1a and related specific comments.

- l.262: '…at least qualitatively capturing major trends in glacier length in many regions': mainly for post-LIA, as this is what you focus on. With some additional analyses and results, as suggested throughout this review could change the 'at least qualitatively' to 'quantitively' ☺

- l.273: 'in almost all cases temperature is the dominant forcing' and l.276: 'suggests negative feedbacks between…on overall glacier geometry changes': indeed, as known from many other studies in which link SMB and climatic forcing is examined (comment 1b + related specific comments). Would make story stronger if you would also explore the link between climate forcing and SMB.

- l.277: 'using dedicated glacier models': real need to have glacier model (vs. only considering the SMB): is to have the lag between SMB forcing and geometry change due to response time. By linking your story to response time (see last suggestion on the 'results' section): reinforce your story line and gives you an additional argument to support the use of OGGM.

- l.282-284: several challenges when considering all glaciers at regional scale. See general comment 5: would be good to at least mention some of the problems that will arise in such a case.

**Bibliography**

Braithwaite, R. J., & Zhang, Y. (2000). Sensitivity of mass balance of five Swiss glaciers to temperature changes assessed by tuning a degree-day model. *Journal of Glaciology*, *46*(152), 7–14. https://doi.org/10.3189/172756500781833511

Hock, R., Bliss, A., Marzeion, B., Giesen, R. H., Hirabayashi, Y., Huss, M., et al. (2019). GlacierMIP – A model intercomparison of global-scale glacier mass-balance models and projections. *Journal of Glaciology*. https://doi.org/10.1017/jog.2019.22

Leclercq, P. W., Oerlemans, J., Basagic, H. J., Bushueva, I., Cook, A. J., & Le Bris, R. (2014). A data set of worldwide glacier length fluctuations. *The Cryosphere*, *8*(2), 659–672. https://doi.org/10.5194/tc-8-659-2014

Lefauconnier, B., Hagen, J. O., Ørbæk, J. B., Melvold, K., & Isaksson, E. (1999). Glacier balance trends in the Kongsfjorden area, western Spitsbergen, Svalbard, in relation to the climate. *Polar Research*, *18*(2), 307–313. https://doi.org/10.1111/j.1751-8369.1999.tb00308.x

Lüthi, M. P. (2014). Little Ice Age climate reconstruction from ensemble reanalysis of Alpine glacier fluctuations. *The Cryosphere*, *8*(2), 639–650. https://doi.org/10.5194/tc-8-639-2014

Oerlemans, J. (2005). Extracting a climate signal from 169 glacier records. *Science (New York, N.Y.)*, *308*(5722), 675–677. https://doi.org/10.1126/science.1107046

Oerlemans, J. (2012). Linear modelling of glacier length fluctuations. *Geografiska Annaler, Series A: Physical Geography*, *94*(2), 183–194. https://doi.org/10.1111/j.1468-0459.2012.00469.x

Oerlemans, J., & Reichert, B. K. (2000). Relating glacier mass balance to meteorological data by using a seasonal sensitivity characteristic. *Journal of Glaciology*, *46*(152), 1–6. https://doi.org/10.3189/172756500781833269

Painter, T. H., Flanner, M. G., Kaser, G., Marzeion, B., VanCuren, R. A., & Abdalati, W. (2013). End of the Little Ice Age in the Alps forced by industrial black carbon. *Proceedings of the National Academy of Sciences of the United States of America*, *110*(38), 15216–15221. https://doi.org/10.1073/pnas.1302570110

Parkes, D., & Marzeion, B. (2018). Twentieth-century contribution to sea-level rise from uncharted glaciers. *Nature*, *563*(7732), 551–554. https://doi.org/10.1038/s41586-018-0687-9

Raper, S. C. ., & Braithwaite, R. J. (2009). Glacier volume response time and its links to climate and topography based on a conceptual model of glacier hypsometry. *The Cryosphere*, *3*, 183–194. https://doi.org/10.5194/tcd-3-243-2009

RGI Consortium, . (2017). *Randolph Glacier Inventory – A Dataset of Global Glacier Outlines: Version 6.0: Technical Report, Global Land Ice Measurements from Space, Colorado, USA. Digital Media*. https://doi.org/10.7265/N5-RGI-60

Sicart, J. E., Hock, R., & Six, D. (2008). Glacier melt, air temperature, and energy balance in different climates: The Bolivian Tropics, the French Alps, and northern Sweden. *Journal of Geophysical Research*, *113*, D24113. https://doi.org/10.1029/2008JD010406

Sigl, M., Abram, N. J., Gabrieli, J., Jenk, T. M., & Osmont, D. (2018). No role for industrial black carbon in forcing 19 th century glacier retreat in the Alps. *The Cryosphere Discussions*. https://doi.org/10.5194/tc-2018-22

Solomina, O. N., Bradley, R. S., Jomelli, V., Geirsdottir, A., Kaufman, D. S., Koch, J., et al. (2016). Glacier fluctuations during the past 2000 years. *Quaternary Science Reviews*, *149*, 61–90. https://doi.org/10.1016/j.quascirev.2016.04.008

Trachsel, M., & Nesje, A. (2015). Modelling annual mass balances of eight Scandinavian glaciers using statistical models. *The Cryosphere*, *9*(4), 1401–1414. https://doi.org/10.5194/tc-9-1401-2015

Zekollari, H., Huss, M., & Farinotti, D. (2020). On the imbalance and response time of glaciers in the European Alps. *Geophysical Research Letters*. https://doi.org/10.1029/2019GL085578

---

## Author Comment (AC1) · 9 Mar 2020

**Response to reviewers**

We thank the three reviewers for their thoughtful and detailed comments, and acknowledge the consensus that the manuscript requires significant changes before publication. A comprehensive revision of the manuscript is underway that responds to the criticism provided, and we demonstrate considerable progress through the process in this response, including production of drafts of all of the additional figures we intend to use. The responses to all 3 reviewers begin with the same general overview and end with the set of new and revised figures which have been produced, but contain a set of specific responses to each review's points in order after the introduction.

The three key threads to the criticism in the reviews, in our eyes, are as follows: 1) the description of both the internal processes of OGGM and the way it is used in our study are ambiguous or insufficient, particularly in relation to the use of GCM data, 2) the scope of the study is not well realised, with conclusions not properly related to the stated aims, and questions raised left unanswered, and 3) the analysis of the results provided is not comprehensive enough, with insufficient quantitative measures of model performance; this also contributes to the insufficient conclusions.

What is significant is that there is no major criticism which requires additional modelling to take place; we have all the data we require, and it is produced in a clear and rigorous way, but the failure of the manuscript lies in inadequately communicating exactly what we have done and what we can determine from the results. With the review comments in mind, a rewrite of the manuscript is underway and could be completed in the classical deadline allowed by the journal for revisions. It includes a much more precise and in-depth description of OGGM and its requisite data inputs, and a considerably expanded set of figures and quantitative measures of model performance aiding a refocused narrative that we believe does a much better job of answering the interesting questions raised in the introduction.

Below we describe the proposed changes and additions to both the text of the manuscript and the data visualisations, and relate them to the specific criticisms they are intended to address. Those changes which have already been made are labelled [I] after the description of the change. Various small corrections that do not warrant special discussion (terminology, sentence structure, etc.) have also already been made, but are not mentioned for the sake of brevity, while other suggested minor text fixes will be made after the more substantial parts of the rewrite are all complete. Drafts of any new or updated figures referenced under 'changes already made' are included at the end of the document.

**Review 1 itemised response**

- **Climatic forcing and calibration**
  The authors explain that they use 6 general circulation models (GCM) to drive OGGM with temperature and precipitation. I suppose that they used monthly fields of surface air temperature and precipitation. Yet no details are given. Is the OGGM-inherent interpolation scheme used for these variables. I think that it is worth to specify the assumed lapse rates. However, my primary concern is the casual description of the climatic forcing. In lack of details, I assume that, after the model calibration, the GCM forcing is directly given to OGGM. Yet the GCM performance will differ around the globe making them more or less suitable for explaining length changes in various regions. In my view, the general practice is to define a common reference period with a climatologically meaningful length and apply the climatic forcing in anomaly mode. In this way, GCM biases in the recent period can be accommodated. This aspect is

**even more relevant in terms of the OGGM calibration. As the authors only refer to the original OGGM publication (Maussion et al., 2019), I assume that the temperature sensitivity parameter ($\mu*$) is automatically calibrated based on the interpolated CRU data set. Correct me if I am wrong. Please specify the calibration time period in OGGM. The automated OGGM calibration implies that when you change to the GCM forcing, the model is not expected to perform well in the recent past, reverting the benefit from the CRU calibration. Changing to anomaly modes will help. The other option is to calibrate $\mu*$ to each GCM. As it stands now, I fear that it is almost impossible to interpret the results.**

This is an issue of weakness in our description of the GCM data, OGGM's usage of climate variables, and the way OGGM performs calibrations. Both OGGM's behaviour and our usage of climate variables are well-defined and consistent, but not communicated well in the reviewed draft of the paper, so we address this issue by providing considerable additional description in the methods section.

We clarify the details of the scaling of climate model data to match 1900-2000 CRU data means, which is the default behaviour of OGGM [I], and describe in detail the nature of the data from the GCMs which is used [I]. We considerably enhance the description of OGGM processes, particularly focused on the way that climate variables are used in the surface mass balance calculation [I]. An explicit description of the calibration of mass balance sensitivity is also provided [I].

- **Representativeness**
  **Please specify how representative the glacier sample that has length observations is for each RGI region. An idea could be to present numbers of hypsometric and glacier-area distribution of the glacier sample as compared to the entire region**

  A new figure (P1) shows the distribution of Leclercq glacier lengths relative to RGI glacier lengths [I]. New figure P2 shows how well represented regions are, compared to each other and through time [I].

- **Regional length changes**
  **You compute the regional relative glacier length changes as the mean of normalised length variations of all glaciers with length observations per RGI region. First, this normalisation overrates the importance of small glaciers. You can see this for Alaska in Fig.1: at several points in time, regional relative length changes exceed 0.5 in less than 10 years which should not reflect the response of the large glaciers. Second, the formation of a mean value is highly susceptible to outliers. Outliers in terms of normalised length changes are expected from the small glaciers in the region because large glacier systems will only show moderate relative length fluctuations. Is it possible to use a more robust measure. I do not think that the median will help, let alone that it will be more informative. Yet a weighting by glacier area might help.**

  The new figure P1 [I] addresses concerns associated with the representativeness of the dataset by making the context of the observations in relation to the true distribution of glacier sizes clear. It is accompanied by a comment in the 'glacier data' portion of the text about the impact of glaciers of different sizes on regional mean length changes [I]. This

incorporates a defence of regional means not weighted by glacier length or area because the impact of small glaciers should be significant when we consider the real size distribution of glaciers [I].

**In addition, you compute the regional mean on an annual basis only considering glaciers with measurements in that year. In many regions you see abrupt step changes in this regional value, which are likely a relict of this strategy. It is therefore very difficult to interpret the regional length change record because they can either arise from the model response or the changing number of considered glaciers from year to year. It would be highly informative to include a plot with the regional length records in addition to the sample size variations through time. An idea for removing the 'spike' behaviour is to assume a linear length change between sparse length observations. In other words, would it make sense to linearly interpolate the observed length record to a yearly time-line for glaciers with irregular sampling of length information. Admittedly, my suggestion is not ideal but it removes the sample-size dependence from the regional length values.**

The suggested behaviour for linearly interpolating Leclercq data points was always used, and this is now made explicit in the text [I], and the suggested addition of a plot of the number of available Leclercq records is included in the form of new figure P2 [I]. The revised figure 2 [I] also touches on the 'spiky' behaviour that can come with the variable number of members in the Leclercq dataset by year, which is discussed in the added text related to this figure.

- **Glacier complexes**
  **In the past, many of the nowadays separated glacier units in the RGI were part of large complexes comprising several glacier branches. As OGGM treats each RGI unit independently, larger glacier complexes in the past are not allowed to form. How important is this fact for the glaciers you are focussing on. The influence of tributaries might well have been important for past length changes even during the period with length observations. Is it possible to consider this effect in OGGM? If not please discuss**

  This problem is not one which can be adequately tackled by OGGM, and is fundamentally an issue of any glacier modelling process which operates on a per-glacier basis. Ice masses being dynamically connected at certain times and not at others cannot be modelled by taking each dynamically separate ice mass at a reference time and modelling it through time in isolation. This is referenced in a new section on 'extensions and limitations'.

- **Calibration strategy**
  **Recently, Eis et al. (2019) presented an alternative for the standard initialisation technique in OGGM. They show that it is important to not only consider present glacier length in the calibration but also the full geometry. In this way, the uncertainties in hindcast simulations can substantially be reduced. As I understand it, you use the standard OGGM spin-up which is not intended to reproduce length changes even in the last century. If there is no length change calibration in this period, why would you expect reliable performance over an entire millennium? Please justify. From my perspective, it is a prerequisite that the approach is calibrated against the length change record (Leclercq et**

**al., 2014) to guarantee a certain reliability over multiple centuries.**

Unfortunately, the model runs were already completed before the Eis et al. (2019) work was available, with our version of OGGM kept constant at the labelled 1.1 release to ensure consistency between runs performed at different times for different regions and different GCMs. As Leclercq et al. (2014) is amongst the best sets of long-term length records for glaciers across many regions, and is still sparse both spatially and temporally, we do not feel that we have sufficient data available to both calibrate to length changes and compare against a separate set of length changes. We do, however, see benefits to considering the performance of a 'naive' model setup, whereby we do not tailor the setup of the model to reproduce one particular variable of interest; by calibrating to a certain variable, we can guarantee that the model produces results that are relatively 'correct' on the largest scale even if it is not the result of significant model skill, which is not the case if we allow the model to work in a way which is agnostic of the variables we will be examining from it. With the results we produce, we are able to find cases where the model performs well and cases where the model performs poorly, and it is the comparison with observed length changes rather than 'correct' lengths over the last 1000 years that is our primary focus.

In the new 'extensions and limitations' section of the discussion, the explanation of the sparse measurements available is reiterated (after featuring in the introduction) and it is made explicit how modelling for a period when there is essentially only one metric with sparse and inconsistent measurements imposes additional restrictions on how we can calibrate the model and what measures of model performance we can produce.

- **Manuscript structure**
  **The 'Methods & Data' section and the 'Results' sections appear as single entities and they lack some structure. For the 'Methods' section, you could present OGGM with some more details on the inherent calibration. For the 'Data' part, you can specify the RGI and the length record with the pre-processing details. Moreover, the climatic forcing could be described in detail. The 'Results' section is a melange with a discussion. I would mention that in the section title. Also try to introduce some sub-divisions.**

  We split the 'methods and data' section into a 'data and model description' section (with subsections for OGGM, the six GCMs used, and glacier data from Leclercq and RGIv6) and an 'experiment description' section on the way our study's specific runs were conducted [I]. We split the existing 'results' section into separate 'results' and 'discussion' sections for readability [I], and provide a heavily modified and extended discussion.

- **Objectives**
  **As already mentioned above, you raise high expectations in your introduction but the conclusions appear rather vague (e.g. abstract, L35, L44, etc.). Therefore, I suggest that you better streamline the manuscript on the conclusions that you are able to draw.**

  The discussion now directly responds to the stated goals and questions raised in the introduction. Two of the new figures, P3 [I] and P4 [I] provide more quantitative measures of the ability of the model to reproduce observed lengths and length changes when driven by each GCM. The new discussion frames results in response to the question of how well the model reproduces 20th century trends under each GCM input when run over longer timescales, and ties in existing data on relative roles of temperature and precipitation.

- **Open dataset**
  **From your description, I appreciate the effort to link the RGI to the length change record from Leclercq et al. (2014). I therefore suggest that you provide a lookup table between the RGI-ID and the ID numbers of the length record, specifying 'positive' and 'best-effort' matches as well as not retrievable entries.**

  A link to the file showing the matching between RGIv6 and Leclercq glaciers is now provided, addressing a request for this information [I].

- **Detailed comments**
  **L35 To what challenges do you refer here? You emphasise this point already in the introduction but you leave it vague here. Please substantiate.**

  We clarify the challenge here, of needing a system that can reach and properly represent stability in a model calibrated for a time period in many cases without near-equilibrium periods for reference, and also discuss the 'theoretical equilibrium finding' behaviour of OGGM's calibration process in the expanded description of surface mass balance and calibration.

  **L107-108 From this sentence, I would have expected that Table 1 comprises number of glacier units and glacierised area per RGI-region. Please add.**

  Number of glaciers are added [I] but areas are not provided, as area is not featured in the Leclercq data, and the only other way to obtain areas is to take more recent area figures from the RGI (when the reference date for Leclercq is 1950) based on the matching we perform. This means that area cannot be used as a comparison between model results and observations, and therefore it is questionable what value its inclusion can bring.

Revised figure 2: the set of modelled glaciers that contribute to the regional average over time now varies to match the set of glaciers that have Leclercq data available for any given year. The intention is to show the impact of the changing number of contributors to the regional means. Generally speaking, where spikes appear across multiple GCM runs in this new figure but are not apparent at the same time in the paper's existing figure 1, this is likely to represent an artefact of the dataset rather than an actual change in modelled glacier lengths.

[Figure]

Proposed new figure P1: The distribution of RGI glaciers vs the distribution of Leclercq glaciers. This is useful for general context on the datasets involved, but also illustrates the considerable bias towards larger-than-average glaciers in the Leclercq dataset, and backs up the claim that is now added; that contrary to the criticism that smaller glaciers in the Leclercq dataset disproportionately affecting normalised regional averages, the Leclercq dataset considerably overrepresents larger glaciers and the larger or more rapid normalised changes that smaller glaciers can experience are likely more representative of the bulk of glaciers.

[Figure]

Proposed new figure P2: Changes to the number of glaciers which contribute to the Leclercq mean by year. This contextualises the potentially 'spiky' nature of the Leclercq averages; where there is a rapid jump in a particular region, it is possible that sudden changes in the mean glacier length in that year are explained as an artefact of the data, rather than representing OGGM outputting rapid changes in glacier length.

[Figure]

Proposed new figure P3: distribution of absolute length errors in 1950. This is part of the effort to address criticisms of the exclusive use of normalised length changes in the submitted draft. We see a moderate bias towards underestimating 1950 length from the CESM-driven runs, and towards overestimating from GISS (despite the mean not reflecting this due to the effect of outliers), and a greater range of length changes generated by the BCC-CSM-driven runs.

[Figure]

Distribution of per-glacier differences between modelled and Leclercq-observation length (absolute)

Proposed new figure P4: Plotting the modelled and observed per-glacier trends over the 20th century (including all glaciers which have 68 or more years in the 20th century covered by the Leclercq timeseries, which represents the point where 90% of glaciers are included). This addresses the issue raised of the glaciers being represented only through regional means. The data shows that the magnitude of observed trends on the scale of individual glaciers is not well modelled by OGGM, and that the differences in how well represented glacier changes are between models using different GCM forcings are small compared to the difference between the modelled changes and the observed changes. The less-than-parity regression line slopes for every model suggest that OGGM is likely to underestimate glacier retreat, especially for larger values of observed retreat. A similar plot for normalised trends shows an almost identical picture, but we choose the absolute trends simply because the required axis scales are less impacted by outliers.

Per-glacier 20th century trends: modelled vs observed (absolute, all glaciers)

---

## Author Comment (AC2) · 10 Mar 2020

**Response to reviewers**

We thank the three reviewers for their thoughtful and detailed comments, and acknowledge the consensus that the manuscript requires significant changes before publication. A comprehensive revision of the manuscript is underway that responds to the criticism provided, and we demonstrate considerable progress through this, including production of drafts of all of the additional figures we intend to use. The responses to all 3 reviewers begin with the same general overview and end with the set of new and revised figures which have been produced, but contain a set of specific responses to each review's points in order after the introduction.

The three key threads to the criticism in the reviews, in our eyes, are as follows: 1) the description of both the internal processes of OGGM and the way it is used in our study are ambiguous or insufficient, particularly in relation to the use of GCM data, 2) the scope of the study is not well realised, with conclusions not properly related to the stated aims, and questions raised left unanswered, and 3) the analysis of the results provided is not comprehensive enough, with insufficient quantitative measures of model performance; this also contributes to the insufficient conclusions.

What is significant is that there is no major criticism which requires additional modelling to take place; we have all the data we require, and it is produced in a clear and rigorous way, but the failure of the manuscript lies in inadequately communicating exactly what we have done and what we can determine from the results. With the review comments in mind, a rewrite of the manuscript is underway and could be completed in the classical deadline allowed by the journal for revisions. It includes a much more precise and in-depth description of OGGM and its requisite data inputs, and a considerably expanded set of figures and quantitative measures of model performance aiding a refocused narrative that we believe does a much better job of answering the interesting questions raised in the introduction.

Below we describe the proposed changes and additions to both the text of the manuscript and the data visualisations, and relate them to the specific criticisms they are intended to address. Those changes which have already been made are labelled [I] after the description of the change. Various small corrections that do not warrant special discussion (terminology, sentence structure, etc.) have also already been made, but are not mentioned for the sake of brevity, while other suggested minor text fixes will be made after the more substantial parts of the rewrite are all complete. Drafts of any new or updated figures referenced under 'changes already made' are included at the end of the document.

**Review 2 itemised response**

- **I find the focus of the paper a bit unclear. From the abstract, I get two objectives, which are obviously linked, but not spelled out very clearly: (i) test whether calibration of the model during the retreat phase is good enough and leads to adequate results also in times of advance or stability, and (ii) identify whether precipitation or temperature anomalies are more responsible for glacier length changes at multi-centennial time scales. I think it might help to use (ii) as the main objective, which would require (i) as an intermediate step.**

  See the response to the point two below ('In the introduction, the attribution...') for more specific discussion on the rationale for refocusing around point (i) as the primary objective and (ii) as a supporting objective. Using the constant-climate runs as a tool for examining the full GCM-driven runs rather than as an end in themselves can help us to understand why the modelled glacier lengths do well or poorly in reconstructing length changes from the observational record.

The discussion now directly responds to the stated goals and questions raised in the introduction by framing results in direct response to the question of how well the model reproduces 20th century trends under each GCM input (see also new figure P4) when run over longer timescales, and tying existing data on relative roles of temperature and precipitation. The conclusions section is rewritten with reference to the bolstered discussion section, and is able to provide more substantive points that link more directly to the introduction.

**This also relates to L38-39: "it is important that these models are examined over time periods where more stable glacier geometries were expected". It could be argued that if a model is not foreseen to be applied in conditions where glaciers are stable or advance, the model does not need to be able to show such behavior (e.g., to my knowledge, the representation of ice geometry change of the model of Huss and Hock, 2015, does not provide for advancing glaciers). It would be good to give some explicit reasons why it is important.**

There are two threads to the response. The first is that the calibration process for OGGM makes an assumption of the existence of equilibrium states and OGGM's ability to keep mass balance close to zero under appropriate conditions. The second is a more generalised point on the philsophy of modelling, whereby it is inherently a deficiency to have a model which cannot reach non-trivial (in this case, not zero length/volume for a glacier) stable conditions if it is intended to model a phenomenon which provably reaches non-trivial stable conditions in reality. Physically, we know that the behaviour of a glacier in response to a change in climate is - with a delay - to reach a new equilibrium, and that is true of any climate change not large enough to either have the glacier entirely melt away or grow limitlessly, and if a model cannot replicate this process of transitions between equilibria, it is somewhat assuming its own conclusions; a model that works only for periods of continual glacier retreat cannot be meaningfully used to predict or hindcast continual retreat of glaciers.

On the first point, we make a short reference to this in the introduction that directs the reader to detailed description of OGGM's mass balance calculation and sensitivity calibration in the methods section [I]. On the second point, we include a version of the argument above, but also recognise that there are cases where the model is expected to work less well (specifically glaciers with terminuses farther along the flowline than they are today, or at any point during the observational record) and later discuss this in the new 'extensions and limitations' section within the discussion.

**In the introduction, the attribution of glacier length change to either precipitation or temperature anomalies seems like an afterthought. I think it would help the paper a lot to restructure with one clear objective (which I think could be this attribution – but the authors may disagree).**

With the additional analysis that has been generated in the process of revising the manuscript, it has become clearer to us that it is necessary to focus on assessing the performance of the model, rather than being able to take the additional step of making the response to individual climate variable forcings the primary subject. This is because we find that despite model skill in reproducing (at least) qualitative regional trends for many regions, the reproduction of the magnitude of per-glacier trends is shown to be poorer, so the matter of model performance under 'normal' is much more complex than just a box to check on

the way to assessment of single-climate-variable runs. In large part this is probably due to the 'chaining' of moving parts that are being examined; not only do we need to test the performance of OGGM running over long timescales (which is already subject to heavy limitations on data to compare against), but we are also implicitly testing the performance of each of the GCMs as a timeseries of climate variables for producing realistic glacier states. To make the constant-climate-variable runs the focus is essentially another link in this 'chain' as we assess the impact of the climate variable isolation simultaneously with OGGM's skill and the GCMs' climate datasets, all with considerable limitations on the observational dataset to compare against.

For this reason, we choose to make the constant-climate-variable runs a tool to examine the full climate runs, rather than the focus of the paper (with all the new analyses that would require), and this purpose is now made explicit, both within the introduction and within the refocused discussion and conclusions sections that have been directly tied to points made and questions raised in the introduction. Points from the above paragraph are incorporated partly into the introduction, and partly into the new 'extensions and limitations' section within the discussion.

- **It probably would be helpful if a bit more is said on the setup of climate forcing in OGGM: the authors are referring to "level 3" preprocessing of OGGM, but don't give any details on how OGGM treats climate model output before application to the mass balance model (this concerns bias correction/anomaly coupling, estimation of solid precipitation, any corrections etc). As it stands now, readers might by surprised by the apparent ability of CMIP5-type models to represent mountain climate conditions accurately enough, which is only half the truth. (see also L166, "as provided", which is not true)**

We considerably enhance the description of OGGM processes, particularly focused on the way that climate variables are used in the surface mass balance calculation [I]. An explicit description of the calibration of mass balance sensitivity is also provided [I] and the justification for the calibration using OGGM's default method is made explicit [I]. We also clarify the details of the scaling of climate model data to 1900-2000 CRU data [I].

- **Regional averages are presented and discussed, and this comes at the relatively high cost of having to find a way how to calculate regional length changes (see discussion around L120-125). It is my impression that the calculation of regional averages is a purely graphical requirement, needed in order to avoid having to inspect 339 individual glacier time series. This points to a more significant problem, which is that the assessment of the model results depends too much on this graphic representation. I would recommend the authors to expand the analysis of results to a more quantitative evaluation, such that the assessment of regional differences depends less on visual inspection of graphs that necessarily are associated with shortcomings (such as the spikes resulting from changes in the observational ensemble).**

There is a significant extent to which the regional grouping of data is a result of questions of how to display the data, though this is in line with many other studies in which RGI regions are considered natural partitions of glacier data. We believe that showing the data on a regional basis is appropriate, given the expectation that glaciers within a region will behave more similarly, and experience greater similarity of climate variability

and trends, than glaciers in different regions. We do however recognise that the importance of the criticism that too much of the assessment of the results rests on the specific representation of regional data we have chosen.

We maintain the region-based display of data in the existing set of figures, but provide a new set of region-agnostic quantitative measures centred on new figures P3 and P4 [I] in order to remove the exclusive reliance on regionalised datasets. We also give additional context to the regional plots by providing a visualisation of the number of glaciers in the Leclercq data per region for each year in new figure P2 [I], and illustrate the impact of the changes in the observational ensemble with a modified figure 2 [I] that, in contrast to figure 1, has the set of glaciers averaged from the modelled dataset vary year-on-year with the set of glaciers available in the Leclercq dataset.

**At least, it might be worth to extract data from the modeled glaciers at the same time as observations exist (adding a third version to Fig. 1 and 2), so that the modeled regional average would have the same spikes as observations if it was perfect. But I think this would not be the optimal solution. The analysis based on linear trends aims in the right direction, but doesn't really relate to the observations, so it is not helping with this issue.**

The revised version of figure 2 [I] serves this function, with the set of glaciers contributing to the model average matching the set of glaciers with available observations from Leclercq, and additional context for these changes is provided by new figure P2 showing the evolution of the size of this set over time for each region.

- **The discussion of Fig. 4-6 would form a nice basis to infer something about the adequacy of the climate models if the comparison of the model results to observations was more quantitative. E.g., it would be possible to quantify how much the glacier model performance is reduced (presumably) if either temperature or precipitation information is withheld, giving further insight into their relevance (and the climate model's ability to represent precipitation and temperature evolution – after all, it is possible that a model's performance increases when one of the two variables is held constant).**

These are great ideas for developing the analysis of constant-climate-variable runs and what they can contribute to our understanding of model performance, but in light of the point above on the paper's focus and concerns over the potential overloading of analyses after the addition of a number of other metrics (see new figures), we would prefer not to add new data analyses or figures that relate specifically to the constant-climate-variable runs unless it is considered necessary for rounding out the paper.

**I also think that the discussion of variances misses the opportunity to say more about physical, climate related causes of regional differences. This is also true for the discussion of relative variances > 1, which basically imply that there is dependency between temperature and precipitation. This is discussed on a technical level (it might be added how the way the solid fraction of precipitation is calculated in OGGM adds to this dependency), but there are also climatological reasons that should be considered here. This should include a discussion of the relevant literature on precipitation vs. temperature influence of glaciers (which is currently almost completely missing). Also, a discussion**

of Marzeion et al. (2014, DOI: 10.5194/tc-8-59-2014), which includes similar experiments, may be helpful.

We give reasons above for not wanting to elevate the material on runs that keep one of the climate variables constant above the core goal of observing and assessing OGGM performance for runs forced with the full GCM data, but we nevertheless expand the discussion in the direction suggested as all of the requisite data is already available.

Within the context of the expanded discussion, the implications of the results from the constant-climate-variable runs are discussed, and they are also related directly to the full GCM climate runs and their performance in reproducing observed length changes. The suggested reference features in this discussion [1].

- **Specific/minor comments/suggestions**

Responses are given only to selected points, with corrections to sentence structure and citations straightforwardly implemented unless otherwise specified.

**L3 (and throughout manuscript): delete "post-" from post-industrial (unless this is a standard term – but to me it sounds like "after the industrial age", which is not what you mean)**

The logic behind the initial terminology was the idea of things happening after the onset of the industrial period, but we appreciate that this is not made sufficiently clear and recognise that the suggested use is an improvement. This is implemented throughout the paper, and any references that are related to the onset of the industrial period rather than to the whole of the subsequent period are made explicit [I].

**L32: OGGM also include precipitation in the calibration**

Along with the comprehensively extended section on OGGM calibration and calculation of mass balance [I], we revise all other references to OGGM's calibration and use of climate data, and where appropriate refer to the material in this new section.

**L35: please spell out some of the additional challenges**

We clarify the challenge here, of needing a system that can reach and properly represent stability in a model calibrated for a time period without near-equilibrium periods for reference, and also discuss the 'theoretical equilibrium finding' behaviour of OGGM's calibration process in the expanded description of surface mass balance and calibration.

**L115-117: "smaller in relative magnitude" yes, but for many applications, the relative magnitude of changes is not that important, but the absolute mass change. So this is maybe not such a big problem.**

This is true, but we do feel that it is sensible to mention. In a paper that was focused on overall volume change, this would likely be a small effect, but as we focus on length changes, the effect is probably not insignificant. New figure P1 [I] provides some context, showing the real disparity in length distribution between Leclercq and the RGI.

**L119-120: it is unclear to me here why a mean regional glacier length estimate is needed? Up to here it seems all comparisons to observations are done on a perglacier-basis (which seems like a better idea to me).**

This point is addressed in the response to the longer 'regional averages...' comment above.

**L 151: from my own experience I appreciate the difficulty of doing the matching, but 38 "not found" glaciers strikes me as a high number. Typically, glaciers with length observations tend to be more "famous" than the average glacier. Might it be worth checking in other data bases (e.g., GLIMS) based on glacier name?**

The method for matching glaciers was actually chosen specifically to avoid matching glaciers which are given the same names but which do not properly match according to an objective standard for glacier location and area data. This is for two reasons: firstly, differences in the way glaciers are partitioned can make even glaciers which are genuinely the same ice mass (or parts of what was once the same ice mass) inappropriate for comparison. Where the same glacier in the RGI and in the Leclercq dataset cannot be reconciled as representing the same dynamically connected ice mass, it is not valuable to compare model results based on the RGI definition to length changes based on the Leclercq definition. Second, where glaciers can be identified across databases by name but either their location or their geometry shows a serious mismatch, it suggests that the quality of the data for that glacier in either or both inventories is not accurate enough, which is a reason not to model that particular glacier based on RGI data and expect a viable comparison with Leclercq observations.

This explanation can be added if it is deemed necessary, but out of a desire for efficiency in an already heavily expanded methods section, we choose not to include it now. The matching method is already described in detail and we also now have the files containing the matched glacier list and the method description linked as an online resource [I].

**L 153: please provide a table linking RGI-ID to Leclerq's database as a supplement (as a service for similar, future studies)**

A link to the file showing the matching between RGIv6 and Leclercq glaciers is now provided, addressing a request for this information [I].

**Fig. 1 & 2: I'm surprised that also the model results are very spiky in some regions. Why is that so? It would also be helpful to use the same vertical axis range in all subplots (even if it cuts off parts of some graphs), since the normalization make the regional comparison easier (and the different axis limits make it harder).**

Spiky model behaviour is typically due to glaciers which, under the model, become very small (but crucially do not disappear) and then in periods of colder/higher-accumulation conditions, have the front of the glacier considered much further along the flowline due to the way OGGM calculates the front of a growing glacier. We discuss this in a new 'extensions and limitations' section within the discussion that examines the weaknesses of OGGM as well as the potential for further use.

We agree that a fixed axis range for the subplots could make for easier comparison between models, but we would prefer not to make this change because we deem the loss of easily-readable information this would result in - cutting off parts of certain graphs, some of which are not spikes that result from modelling issues but actual large relative changes changes, and 'squashing' the output of different GCM runs together in regions which currently use smaller vertical axis ranges (e.g. Southern Andes) - to be greater than the increase in readability from a region comparison perspective.

**L182-186/Fig 1&2: I don't really understand what makes regions 14, 15 and 18 stand out from the other regions? Again, I think a more quantitative comparison of the model ensemble to the observation ensemble would be very helpful.**

The description of the individual features of these regions is not particularly effective, so this section is removed [I] as part of the rewritten results section with a more quantitative focus, including the additional information made available by the new graphs P3 and P4 [I].

**L189: not sure what is meant by "stratified", so I also don't understand the following argument**

We now use 'when the gaps between the output for different GCMs are large compared to the internal variability of a single output' instead of 'When the model results are highly stratified' [I]. Use of the term 'stratified' was overly reliant on how the lines look on the graph, while the new phrasing refers to features of the data.

**L199 and following: would it make sense to do this analysis first for each glacier in the region, and then build the regional mean? This could result in an indication how robust the estimation of the "inflection" year is, and also give some more information on intra-regional variability of the glaciers' behavior.**

The issues that exist with performing a split regression for each glacier and then combining them are 1) that individual glaciers are more sensitive to variability than a mean of a larger set of glaciers (see also the next reviewer point on the sensitivity of the inflection year), and 2) that there is no clear way to combine these sets of split regressions to form a useful mean. It is easily possible to determine a mean year of inflection, but then there is no intuitive idea of a mean slope before and after this point. Nevertheless, we do have an increased focus on per-glacier rather than per-region data in the form of new figure P4 [I] directly, and indirectly through quantitative measures which take per-glacier distributions rather than focusing on data from regional mean outputs (including new figure P3 [I]).

**L213-125: I find this argument a bit weak, given that the "inflection" year is probably very sensitive to short-term interannual variability in the climate time series**

As short-term variability is always present within the timeseries and we overwhelmingly see the 'inflection' year in a relatively short window around the onset of warming in the industrial period, it does seem that it is longer-term trends which dominate the determination of the inflection year. The sensitivity to short-term changes is also mitigated somewhat by performing the split regression on regional means rather than individual glaciers, so impacts of glaciers in situations prone to rapid advance or retreat in response to shortterm changes are somewhat dampened. We do, however, agree that the idea of linking the inflection year to the year that recent retreat begins, even though we do not identify the two, may be overstating what can be determined from this data, so this point is rewritten to avoid speculation about the year retreat begins determining the overall size of retreat. Instead we comment quantitatively on the relationship (or lack thereof) between the inflection year and the slope of the 2nd part of the regression (replacing the purely quantititative expression).

**Fig. 5: was the non-zero melting threshold temperature of OGGM taken into account when calculating the temperature time series shown here (also, in L226, it says "degree-days" – is it really days, or months)?**

The same melt threshold as used in OGGM forms the basis for the degree-months. With the new detail provided in the methods section on the processes in OGGM, and particular focus on surface mass balance [I], all references to OGGM method have been checked, and if needed made more specific and directly referenced the new detailed description where necessary [I].

**L262-263: it is not only OGGM, but (probably at least as much) the GCMs that are responsible for the level of agreement.**

This specific statement is removed with the restructuring of the discussion and conclusions to directly and consistently reference the questions and aims raised in the introduction. However, we also make reference to the difficulty of conclusively determining which of OGGM and the GCM data is responsible for features we see in the outputs due to the 'chaining' of models - the GCM itself, which then feeds data into OGGM - in the new 'limitations and extensions' section.

Revised figure 2: the set of modelled glaciers that contribute to the regional average over time now varies to match the set of glaciers that have Leclercq data available for any given year. The intention is to show the impact of the changing number of contributors to the regional means. Generally speaking, where spikes appear across multiple GCM runs in this new figure but are not apparent at the same time in the paper's existing figure 1, this is likely to represent an artefact of the dataset rather than an actual change in modelled glacier lengths.

[Figure]

Proposed new figure P1: The distribution of RGI glaciers vs the distribution of Leclercq glaciers. This is useful for general context on the datasets involved, but also illustrates the considerable bias towards larger-than-average glaciers in the Leclercq dataset, and backs up the claim that is now added; that contrary to the criticism that smaller glaciers in the Leclercq dataset disproportionately affecting normalised regional averages, the Leclercq dataset considerably overrepresents larger glaciers and the larger or more rapid normalised changes that smaller glaciers can experience are likely more representative of the bulk of glaciers.

[Figure]

Proposed new figure P2: Changes to the number of glaciers which contribute to the Leclercq mean by year. This contextualises the potentially 'spiky' nature of the Leclercq averages; where there is a rapid jump in a particular region, it is possible that sudden changes in the mean glacier length in that year are explained as an artefact of the data, rather than representing OGGM outputting rapid changes in glacier length.

[Figure]

Proposed new figure P3: distribution of absolute length errors in 1950. This is part of the effort to address criticisms of the exclusive use of normalised length changes in the submitted draft. We see a moderate bias towards underestimating 1950 length from the CESM-driven runs, and towards overestimating from GISS (despite the mean not reflecting this due to the effect of outliers), and a greater range of length changes generated by the BCC-CSM-driven runs.

Distribution of per-glacier differences between modelled and Leclercq-observation length (absolute)

[Figure]

Proposed new figure P4: Plotting the modelled and observed per-glacier trends over the 20th century (including all glaciers which have 68 or more years in the 20th century covered by the Leclercq timeseries, which represents the point where 90% of glaciers are included). This addresses the issue raised of the glaciers being represented only through regional means. The data shows that the magnitude of observed trends on the scale of individual glaciers is not well modelled by OGGM, and that the differences in how well represented glacier changes are between models using different GCM forcings are small compared to the difference between the modelled changes and the observed changes. The less-than-parity regression line slopes for every model suggest that OGGM is likely to underestimate glacier retreat, especially for larger values of observed retreat. A similar plot for normalised trends shows an almost identical picture, but we choose the absolute trends simply because the required axis scales are less impacted by outliers.

Per-glacier 20th century trends: modelled vs observed (absolute, all glaciers)

---

## Author Comment (AC3) · 10 Mar 2020

**Response to reviewers**

We thank the three reviewers for their thoughtful and detailed comments, and acknowledge the consensus that the manuscript requires significant changes before publication. A comprehensive revision of the manuscript is underway that responds to the criticism provided, and we demonstrate considerable progress through this, including production of drafts of all of the additional figures we intend to use. The responses to all 3 reviewers begin with the same general overview and end with the set of new and revised figures which have been produced, but contain a set of specific responses to each review's points in order after the introduction.

The three key threads to the criticism in the reviews, in our eyes, are as follows: 1) the description of both the internal processes of OGGM and the way it is used in our study are ambiguous or insufficient, particularly in relation to the use of GCM data, 2) the scope of the study is not well realised, with conclusions not properly related to the stated aims, and questions raised left unanswered, and 3) the analysis of the results provided is not comprehensive enough, with insufficient quantitative measures of model performance; this also contributes to the insufficient conclusions.

What is significant is that there is no major criticism which requires additional modelling to take place; we have all the data we require, and it is produced in a clear and rigorous way, but the failure of the manuscript lies in inadequately communicating exactly what we have done and what we can determine from the results. With the review comments in mind, a rewrite of the manuscript is underway and could be completed in the classical deadline allowed by the journal for revisions. It includes a much more precise and in-depth description of OGGM and its requisite data inputs, and a considerably expanded set of figures and quantitative measures of model performance aiding a refocused narrative that we believe does a much better job of answering the interesting questions raised in the introduction.

Below we describe the proposed changes and additions to both the text of the manuscript and the data visualisations, and relate them to the specific criticisms they are intended to address. Those changes which have already been made are labelled [I] after the description of the change. Various small corrections that do not warrant special discussion (terminology, sentence structure, etc.) have also already been made, but are not mentioned for the sake of brevity, while other suggested minor text fixes will be made after the more substantial parts of the rewrite are all complete. Drafts of any new or updated figures referenced under 'changes already made' are included at the end of the document.

**Review 3 itemised response**

- **1. At this point, many statements a bit vague and the analysis presented are not very in-depth. Additional information (mainly quantitative information) and analyses will be required to really make point that the authors are able to use OGGM to closely reproduce past glacier changes. Two main points here: a. You chose to mainly focus on relative length changes at regional scales and explain why you present the results in such a fashion. This is definitely fine, but I do not think this should impede you from also giving some results in absolute values and for individually modelled glaciers. This is just a matter of showing the results differently and does not require any additional runs/simulations. More specifically, it would be interesting to see how the model is able to reproduce the present-day glaciers: i.e. are the glaciers that you obtain at the end of your simulation (around 2000) close to the observed ones? So far, you argue that it is important to look at the model for periods when glaciers were more stable (e.g. l.38-40), but this should not stop you from also considering**

**the model performance for the recent past and its capability to closely reproduce the present-day glacier. An example: take a glacier for which the relative length change is well reproduced over the last 300 years, but where your modelled present-day length is 8 km vs. an observed length of 20 km... This means that your glacier was also 2.5 times too long 300 years ago (everywhere between 300 years ago and now): would you argue that the model does a good job at representing the changes here? Would be good to have a figure (e.g. in suppl. Mat.) with on the x-axis the observed glacier length at present-day, on the y-axis the modelled present-day glacier length (after transient run) and having every individual glacier plotted in this (for all regions together; could do this for one climate model)**

Additional quantitative measures form the bulk of the new analysis in the revised version of the paper. However, an initial point should be clarified, regarding the comparison of length changes vs the comparison of length in a given year. While the GCM data is scaled in such a way that the 1900-2000 mean precipitation and temperature are made to match the mean over this period in the CRU 1900-2000 dataset (on which the mass balance sensitivity is calibrated), we recognise that there will be overall biases in each GCM in the rest of the last millennium. This will see different sizes for modelled glaciers at the start of the 20th century when driven by different GCMs, which will result in different lengths during the 20th century regardless of matching climate variable means. Geometry responses to climate are also non-linear, so the matching climate variable means for 1900-2000 do not mean matching aggregated surface mass balance. Looking at changes (including absolute changes) allows us to focus on changes in climate over time rather than having to account for mean climate differences and changes over time simultaneously. We also note that due to the calibration of GCM climate variables to match CRU 1900-2000 means, this may have some effect of an artificial convergence of 20th century mean absolute glacier lengths (equilibria will be similar for climate datasets with the same temperature and precipitation means), but the same effect does not apply to patterns of change over the 20th century.

Three new figures are created (two of which are shown below) which deal with each of the concerns here; regional aggregation, absolute length comparison, and performance metrics for individual glaciers. New figure P3 [I] looks at the distribution of absolute differences between observed and modelled 1950 lengths - while the focus is on changes, having this context on whether models tend to over- or underestimate glacier lengths in the year other figures are normalised to is important. The new figure P4 [I] gives a scatter plot of 20th century length changes in the model vs the observations, as a quantitiative measure of the performance of the model for each glacier in a way which is not aggregated regionally. Finally, the exact figure suggested at the end of the comment is generated, but not included here; the caveat with this figure is that present-day (i.e. end-of-modelling-period) modelled lengths cannot be compared particularly well with Leclercq observations (see drop off of the number of Leclercq records available in many regions towards the end of the 20th century), so it is necessary for the observed 'present day' glaciers to instead be taken from the RGI, which also provides the geometry for initialising glaciers before the model run. This inconsistency in the nature of the length records used for comparison can reduce the clarity of the results, so we suggest this figure should appear in the supplementary material rather than the main part of the paper. A review of the justification in the text for a focus on a) changes and b) normalised lengths is conducted, incorporating the clarification above.

**b. Role of the SMB. You barely mention the SMB component of the model,**

which I found surprising, given that this is the main driver for the glacier behavior (the dynamics then translate your SMB forcing – with a lag due to the response time – to a length change). For instance, when considering the role of temperature vs. precipitation forcing, it would be highly relevant to describe how much these components affect the modelled SMB (with quantifiable information). Many studies have provided insights in the role of temperature vs. precipitation forcing for the SMB (e.g. Lefauconnier et al., 1999; Braithwaite & Zhang, 2000; Oerlemans & Reichert, 2000; Sicart et al., 2008; Trachsel & Nesje, 2015), and often found that the temperature is the main driver. Your finding that temperature is the main driver directly results from the calculated SMB, which is far more sensitive to temperature changes than it is to precipitation changes. Would be really nice if you could show some of the calculated SMBs and perform some basic sensitivity tests (e.g. what happens with SMB when forced with +1 degC, -1 degC, +20% precipitation,…etc.)**

The additional detail on SMB is important to include, but we do have significant concerns about whether the suggested additional synthetic runs would actually add to the paper. Our specific intent is to examine the way OGGM reproduces glacier lengths under specific GCM climate datasets, and the runs with constant temperature or constant precipitation are ways to determine which of these explains most of the reproduced glacier changes. The suggested runs do not seem to serve this specific purpose; rather they are interesting pieces of general glacier model testing, but do not relate to the aims of this particular paper.

We do, however, include the suggested references for context on the variance explained by the constant-climate runs. In general, we considerably enhance the description of OGGM processes, particularly focused on the way that climate variables are used in the surface mass balance calculation [I]. An explicit description of the calibration of mass balance sensitivity is also provided [I] and the justification for the calibration using OGGM's default method is made explicit [I]. We also clarify the details of the scaling of climate model data to 1900-2000 CRU data [I].

- **2. Title is a bit misleading: when reading 'regional' glacier length changes, I would expect that an entire region is considered. However, glaciers from various regions are selected, which in every case represent only a very small subset of all glaciers in this region. Suggest reformulating this, which could be done by simply omitting 'regional'. Or should rather mention something like: 'in various regions around the world'**

It is of course true that specific subsets of glaciers are used, but we do feel the term 'regional' is appropriate because we are clear about the number and distribution of glaciers used (see new figures P1 and P2), and because there is no cherrypicking of glaciers; we simply use all the glaciers that have available length records in one consistent format. See also the response to general comment 5 on the issues that face all modelling efforts that try to operate on a per-glacier basis; we believe that using an available dataset which is separated into regions and describing the outcomes as 'regional' is consistent with established glacier modelling literature, even when the number of glaciers is significantly larger than those used here. We are just so lacking in long-term direct observations of glaciers that our approach is as 'regional' as we can be.

A short comment on regional representativity is added, referencing the new figure P1 on

Leclercq length distribution relative to the whole of the RGI distribution, with context from the relative scarcity of longer term data and the potential incompleteness of even modern inventories [I]. The manuscript is not yet renamed, but we are open to the possibility of renaming it if it is determined the 'regional' label is considered genuinely misleading.

- **3. Almost all the 'action' occurs in the pre-frontal glacier region (compared to the present-day ice cover): how well is OGGM able to handle this? More information is needed about how the flowlines are defined here, how the cross shape is parameterized,... etc. This information is lacking at the moment.**

It is difficult to get a hugely accurate picture of how well OGGM handles this as it's difficult to determine how much of a historical (longer than present day) glacier state generated by OGGM is due to the way that glacier geometry for flowlines extended beyond their current bounds is handled and how much is due to differences in glacier evolution between the model and reality. Naturally examples of glaciers with periods of observed advance are relatively rare, and where they exist are often the result of unique processes like surging which OGGM does not handle well. The greatest uncertainty lies in the calculation of the hypothetical flowline extending from the initial glacier geometry that is used if the glacier advances; the width and cross section of the glacier are all handled identically for glacier extent beyond modern bounds, using the total ice mass at each elevation band to determine the dimensions, but the flowline for the initialised glacier is based on an algorithm applied to observed geometry while the below-glacier-terminus flowline comes from an interation on gradient from the end of the glacier. This method can struggle to deal with cases where glacier dynamics may cause the glacier to flow in ways which are not necessarily in the direction of steepest local gradient (e.g. heading over a lip of rock that is in the direction of existing ice flow).

The above commentary is included in the 'extensions and limitations' section [I] and referenced where there is a mention of glaciers which extend beyond their modern boundaries. The same section also lists mitigating factors - such as the feedback of tongue elevation on overall mass change (and therefore length change) - which suggest that errors in flowlines beyond the modern tongue of the glacier should not cause divergent progressions of glacier evolution between reality and the model.

- **4. I generally found the manuscript relatively easy to follow and found the figures to be simple, but clear, which is very nice. At several occasions I did however get lost in long sentences (often multiple brackets are being used...) and had to read through these several times before getting the meaning of the sentence. I therefore suggest reducing the use of brackets, and splitting up long sentences where possible. Examples are provided in the 'specific comments' section below.**

The issue of multiple brackets was mentioned a couple of times, and the paper has gone through a pass to restructure the sentences where they appear in order to enhance readability and remove the need for excessively complex formulations. We do not list all the responses to individual sentence structure concerns below; if the problem is purely a matter of fixing the text and no meaning needs to be clarified, the specific review comment is not listed and can be assumed fixed with a simple rewrite.

- **5. You describe this study as being a kind of first attempt to reproduce past length changes with a flowline model for glaciers in many different regions and**

suggest that this would open the door to regional scale applications. I agree with the former, but have some doubts about the possibility to fully extend this to regional scales. What about glaciers that are now separate ice bodies but used to be connected? What about glaciers that disappeared by now but may have existed before (whether as separate ice bodies or tributaries to present-day glaciers) – a field in which Parkes himself authored an important study (Parkes & Marzeion, 2018). I think it would be fair to also mention these issues/challenges in your conclusion, were you provide an outlook (last sentence of the manuscript).

These concerns are entirely valid, but are fairly uniformly difficult for all large-scale glacier modelling efforts which separately model each individual ice mass. We do not expect these impacts to be greater (or indeed lesser) in the extension to modelling entire regions (which is already underway), and it therefore should not make these efforts impossible, at least by the standards of existing regional modelling. The challenges of modelling that attempts to be comprehensive, rather than restricted to a limit set of glaciers, should be discussed, but not in the sense of it being prohibitively difficult.

The added 'extensions and limitations' section is where new problems that come from modelling regions with tens of thousands of glaciers, along with problems due to uncertain on total glacier numbers or on ice masses that vary between separate and contiguous through time, are discussed.

- **Specific comments**

Responses are given only to selected points, with corrections to sentence structure and citations straightforwardly implemented unless otherwise specified.

**l.13-16: statement is again rather qualitative here: could you provide concrete numbers that support this statement?**

In light of the enhanced quantitative measures and expanded discussion separated into specific topics - see answer to general comment 1a - the end of the abstract is entirely rewritten, rather than keeping existing points and adding data.

**l.22: 'direct observations of glacier geometry': what do you consider being a direct observation?**

We consider direct observations to be any observation which makes a contemporaneous measurement of the ice mass. This includes satellite mapping, for example, but does not include reconstructions of historical glacier extent from things like moraines.

**l.25-26: 'though even this is likely a significant underestimate (Parkes and Marzeion, 2018)'. Well, the number of glaciers is simply subjective, as it is related to the threshold that is used to decide whether a glacier is mapped (outlined) in the Randolph Glacier Inventory (RGI Consortium, 2017). We know that the number would be higher if smaller ice bodies would also be considered, so would not refer to this as an 'underestimation' here.**

I think this is a misunderstanding of what Parkes and Marzeion (2018) says. It is explicit in not lowering the threshold for the surface area of what can be considered a glacier, restricting the upscaling to glaciers above the cutoff threshold used in the RGI. That there is an underestimation of terrestrial ice masses due to the partly arbitrary nature of this cutoff has some potential impact, but that is not the underestimation being referenced here. The referenced paper suggests significant underestimations of glaciers even an order of magnitude greater area than the RGI cutoff.

**l.32: 'by default calibrates the glacier sensitivity to local temperatures based on CRU data...': what criterion is used for calibration? You mention that CRU data is used, but what do you (try to) match in the calibration procedure? i.e. what is the target? (e.g. measured SMB, geodetic mass balance,...etc.)**

We considerably enhance the description of OGGM processes, particularly focused on the way that climate variables are used in the surface mass balance calculation [I]. An explicit description of the calibration of mass balance sensitivity is also provided [I] and the justification for the calibration using OGGM's default method is made explicit [I]. We also clarify the details of the scaling of climate model data to 1900-2000 CRU data [I].

**l.39: 'we expect that...': strange formulation. You expect smaller and globally less consistent temperature trends? Based on what? Or is this just what the reconstructions suggest? Would rather formulate in lines of 'Studies/Observations suggest that temperature trends were smaller and globally less consistent...'**

This is based on the cited studues (Neukom et al. 2019; PAGES 2k Consortium 2013, 2017) on the lack of evidence for globally consistent temperature trends over the last 2 millennia outside of recent warming in the industrial period. The suggested change to refer to 'studies' is made [I].

**l.45: when mentioning the glacier length changes, could make link with observed changes from Leclercq et al. (2014) (which you use later, but reader does not know at this point) and Solomina et al. (2016) in which the literature on glacier geometry changes over the last 2000 years is summarized.**

The earlier reference to introduce Leclercq et al. (2014) is added, and the new reference to Solomina et al. (2016) is now included.

**l.45: 'we cannot compare': you as authors? Or the literature in general?**

The literature in general. We replace the statement 'Due to the limits on the observational data for comparison, we cannot compare model results with an accurate representation of pre-industrial relative glacier lengths' with the more specific 'The small number of available length records which extend back further than 150-200 years (reference to new figure P2) heavily limits any possible comparison of model results with observed pre-industrial glaciers lengths'.

**l.46-47: focus is on the transition from more stable pre-LIA to retreat. This is a complex matter – and many studies have tried to shed a light on this and came up with several possible mechanisms to explain the timing of this transition**

(e.g. Painter et al., 2013; Lüthi, 2014; Sigl et al., 2018). Would be surprising that your relatively simple setup (with temperature and precipitation forcing only) is able to simulate the right timing (as it is generally known that retreat starts before a real increase in temperatures is observed). Would be good if you could provide a few words of explanation on this.**

We include the suggested references on how the matter has a complexity that can cause significant challenges (and which serves as a 'hard' test of the model reproducing observed glacier timeseries features).

**l.48-50: observations - any reference for this?**

We add a reference here to the Leclercq et al. paper once again, and to the Solomina et al. paper suggested as an addition for the first comment about line 45.

**l.50-52: agree, European glaciers are indeed not representative for worldwide glacier fluctuations. This is clear from recent Glacier Model Intercomparison Project (GlacierMIP), in which a strong contrasting behavior between the evolution of glaciers in various regions is highlighted (Hock et al., 2019).**

The mentioned reference is added.

**l.68: '...comparisons of between models and differences between regions': which models are you referring to here? Glacier and/or climate models?**

The statement is modified to '...comparisons between runs driven by diferent GCMs and differences between regions' to make this clear [I]. There are several instances of ambiguity in whether 'model' refers to a GCM or to OGGM, so every references to 'model', 'models', 'modelled' is checked to ensure it is explicit what is being referred to.

**l.74-75: strange description for OGGM: model for glacier dynamics that accounts for geometry and ice dynamics. Ok, but is also really a model in which SMB is coupled to glacier dynamics to simulate the temporal evolution of glaciers. Would suggest already mentioning the SMB here, and giving more information about the SMB in general in the following sentences, as this is the main driver for your changes over the past centuries... (see general comment 1b)**

We considerably enhance the description of OGGM processes, particularly focused on the way that climate variables are used in the surface mass balance calculation [I]. An explicit description of the calibration of mass balance sensitivity is also provided [I] and the justification for the calibration using OGGM's default method is made explicit [I]. We also clarify the details of the scaling of climate model data to 1900-2000 CRU data [I].

**l.81-84: you use an uncalibrated version of the OGGM model. What are the implications for the modelled glacier geometries at present-day (i.e. after several centuries of transient run): do you end up having a realistic glacier shape? Would be surprising that this can be obtained without any calibration and by just taking the model as is: see also general comment 1a: would be good if**

**you could show the modelled present-day geometry (after transient run) vs. observed (and thus not only rely on relative changes).**

OGGM is used with its default calibration, which finds sensitivity values for SMB based on CRU 20th century climate data and a set of observed glacier SMB values. It is however not calibrated in any way to the data used in this experiment - either to find sensitivity values which are specific to each of the GCM climate datasets used or to match the lengths from the Leclercq data at any point. See preceeding point for the information added on OGGM processes that makes this explicit. See end of response to general comment 1a for comparison of modern observations and end-of-run state of (a specific aspect of) glacier geometry.

**Would need information about pre-frontal area and how you treat glacier changes here. See general comment 3 for more information**

See general comment 3 for detail.

**l.112-117: in your explanation, you link the glacier response time to its size. Is however not the case for many cases / regions (Raper & Braithwaite, 2009; Oerlemans, 2012; Zekollari et al., 2020), and quite often the main driving mechanism for the glacier response time is the surface slope. Could simply reformulate this by saying that the glaciers you consider are typically large and relatively gently sloping glaciers and that these may not be representative for all glaciers in the region when it comes to their response time, as this is driven by a combination of glacier-specific factors.**

This is a good reason to diversify the explanation for response time expectations. We include the suggested references and adopt the idea of discussing the likely gentler slopes in a set of glaciers with a distribution which trends much larger than the distribution of glaciers recorded in the RGI as a whole (new figure P1 [I]).

**l.118-132: OK to have regional values and relative changes, but also need to show your results in absolute values and for individual glaciers. Does not require additional simulations, just a different and elucidating way of looking at results. See comment 1a for more info.**

See response to general comment 1a for list of changes made to provide additonal results that address the need for absolute values and individual glacier data.

**l.139: the area given in the Leclercq dataset: area at which time period? Guess this depends on the region/glacier considered? Would be good if could give indication.**

The Leclercq reference date for area varies and is not always clear. Typically we expect the year to be beyond 1950 - given that every glacier had a record of length observations which extends beyond 1950 - and to be more likely to bias later than this where the area measurement is a product of larger-scale earth observation techniques. The criteria we use are developed with the expectation that there can be ~50 years difference between the Leclercq area observation date and the RGIv6 observation date.

We now make it explicit that the year for the Leclercq area measurement is variable (and not fixed to 1950 like the Leclercq reference length), and add the sentence on the expectations for the criteria [I].

**l.146: 'time of the Leclercq measurement': when is this?**

This refers to the same time as the preceeding point. This is now clarified as 'the year of the Leclercq area measurement' here, and specificity is added about this year as described above.

**l.166-168: you explain that sometimes not fully equilibrated after 300-yr spin-up: if this is the case, why do you simply not consider a longer spinup (¿ glacier response time) of e.g. 1000 years? Seems that in any case an initial adjustment will occur, because glaciers are never entirely in steady state, and you use this as a starting point. This is OK, but could reformulate this.**

This explanation was not sufficient, so we revise it to the following:

'In cases where glaciers are still undergoing significant adjustment to a new equilibrium (e.g. in region 14) after several hundred years of spin-up and the early part of the main run, this is good evidence that in a 1000 year period, responses to trends in the forcing climate variables may not actually be shown in the OGGM output. This does not invalidate the glacier model output, but the evidence of continuing adjustment leftover from the spin-up being shown in the output rather than being removed with an arbitrarily long spin-up might inform the interpretation of the rest of the timeseries. It is also the case that where continued adjustment is significant after several hundred years, the magnitude of the length changes is typically large, and in these cases adding additional spin-up centuries will not fix the fact that the modelled glacier is diverging from the size of the observed glacier. We choose to maintain the 300-year spin-up for the sake of consistency as well as these reasons.' [I]

**l.180: 'models underestimate the retreat shown in the observations': how come? Maybe also role for other factors not accounted for in your model: e.g. role of aerosols (global dimming) and other mechanisms?**

It is not possible to quantify the potential effect of unrepresented processes in OGGM from the information in our model runs, but we do now discuss model limitations at length in the 'extensions and limitations' discussion section, which is referenced wherever we discuss discrepancies between observed and modelled length changes which are outside the range of results provided by the 6 GCMs.

**l.187: '. . . use of normalised glacier lengths removes the ability to tell which. . . ': well, you can simply also additionally give your results as non-normalised glacier lengths. Need to do this to increase insights in your results and capability of OGGM to reproduce the present-day glaciers (after multi-century transient run): see general comment 1a.**

See response to general comment 1a for comprehensive comment on absolute and normalised glacier lengths.

**l.213-215: 'This suggests that differences in total post-industrial retreat are more influenced by differences in when the retreat starts than by...': OK, from the modelling perspective. Is this also the case when considering the observations?**

The intention here was to talk about differences between GCMs (for the same region), so with one observational timeseries for the region, there isn't a comparison between observations to make. We update this to refer specifically to '...differences in total recent retreat between GCM runs for a region are...' to make sure this is clear.

**l.223: '... we do not examine the patterns of pre-industrial length change on a per-region basis': why not? Would be interesting to examine this and this aspect would add novelty to the paper. In the end, the manuscript almost solely focuses on post-industrial time period (although not clear if modelled absolute glacier length changes are realistic, see comment 1a) and when the retreat starts here, although the simulations cover a millennial timescale.**

Ultimately it is matter of a lack of data for comparison. It would be possible to compare models to each other, but there are only a handful of pre-industrial length change observations available (see new figure P2), making comparison to observations largely impossible. This prevents any discussion of how well OGGM performs for each GCM in the pre-industrial period as it restricts commentary to whether different GCMs produce similar or different results, rather than how well they are able to reproduce observations. A more robust defence of this idea along these lines is added to the text.

**l.226: Results shown only for 1 climate model: definitely ok, but would be good if you could argue / give a reason why IPSL is chosen (without this explanation this seems rather arbitrary).**

A line is added on IPSL being chosen due to the apparent early start of recent retreat compared to others (with the expectation that this makes differences between full GCM runs and constant-climate runs more obvious) [I]. The results for all models are shown in the supplementary material, and relatively arbitrary choice is an inevitability unless we show 12 large figures for constant-climate runs, or condense the information heavily.

**l.224-256: in general a lot of rather qualitative statements are made at the end of your result section, which does not leave the reader with real take-away messages: i.e. what should one remember when reading this section? Two suggestions for topics to focus on / include in your discussion:**
**- I think it is important to focus more on the SMB here and link this to other studies in which the SMB and its sensitivity to temperature and precipitation changes are described + show this for OGGM's SMB component used in this study (see general comment 1b + comment on l.74-75).**

See responses to general comments 1a and 1b for full story on enhanced quantitative aspects and SMB.

**What about the response time? This is not really mentioned in your story, but in the end this plays an important role, as the response time will explain**

the lag between the change in SMB (the effect signal of your climate input) and the change in glacier geometry (the glacier length you consider for model evaluation). Would put more emphasize on response time in this section and e.g. explain inter-region and inter-glacier differences in the timing of the post-industrial retreat and how this can be linked to response time.

This point actually gets primarily referenced in the 'extensions and limitations' section of the new discussion, given the fact we know the response time does introduce an amount of lag to the responsiveness of length changes, but we cannot directly ascribe a single response time value to a glacier that is invariate through geometry changes over time and through different types and timescales of climate variation. It is now referenced as a potential explanatory factor, but the potential for introducing uncertainty is given more focus.

**l.261: 'this observed retreat is within the range of the modelled retreats': difficult to judge as only relative changes are given and over regional scales. See comment 1a and related specific comments**

See response to general comment 1a for comprehensive comment on choice of comparisons.

**l.262: '...at least qualitatively capturing major trends in glacier length in many regions': mainly for postLIA, as this is what you focus on. With some additional analyses and results, as suggested throughout this review could change the 'at least qualitatively' to 'quantitively'**

See response to general comment 1 for general material on quantitative analyses.

Comments on 'qualitative' vs 'quantitative' comparisons are comprehensively reviewed in light of the new analyses discussed elsewhere. Here specifically we also change 'major trends in glacier length during the period of observational record' as this is what actually determines the time period in which we can assess the modelled trends.

**l.273: 'in almost all cases temperature is the dominant forcing' and l.276: 'suggests negative feedbacks between...on overall glacier geometry changes': indeed, as known from many other studies in which link SMB and climatic forcing is examined (comment 1b + related specific comments). Would make story stronger if you would also explore the link between climate forcing and SMB.**

See response to general comment 1b for in-depth comment.

**l.277: 'using dedicated glacier models': real need to have glacier model (vs. only considering the SMB): is to have the lag between SMB forcing and geometry change due to response time. By linking your story to response time (see last suggestion on the 'results' section): reinforce your story line and gives you an additional argument to support the use of OGGM.**

See response to 'What about response time?' above.

**l.282-284: several challenges when considering all glaciers at regional scale.**

**See general comment 5: would be good to at least mention some of the problems that will arise in such a case.**

See response to general comment 5. The 'extensions and limitations' section goes into detail on what additional challenges there are in extending our approach to the scale of entire regions as defined by recent inventories.

Revised figure 2: the set of modelled glaciers that contribute to the regional average over time now varies to match the set of glaciers that have Leclercq data available for any given year. The intention is to show the impact of the changing number of contributors to the regional means. Generally speaking, where spikes appear across multiple GCM runs in this new figure but are not apparent at the same time in the paper's existing figure 1, this is likely to represent an artefact of the dataset rather than an actual change in modelled glacier lengths.

[Figure]

Proposed new figure P1: The distribution of RGI glaciers vs the distribution of Leclercq glaciers. This is useful for general context on the datasets involved, but also illustrates the considerable bias towards larger-than-average glaciers in the Leclercq dataset, and backs up the claim that is now added; that contrary to the criticism that smaller glaciers in the Leclercq dataset disproportionately affecting normalised regional averages, the Leclercq dataset considerably overrepresents larger glaciers and the larger or more rapid normalised changes that smaller glaciers can experience are likely more representative of the bulk of glaciers.

[Figure]

Proposed new figure P2: Changes to the number of glaciers which contribute to the Leclercq mean by year. This contextualises the potentially 'spiky' nature of the Leclercq averages; where there is a rapid jump in a particular region, it is possible that sudden changes in the mean glacier length in that year are explained as an artefact of the data, rather than representing OGGM outputting rapid changes in glacier length.

[Figure]

Proposed new figure P3: distribution of absolute length errors in 1950. This is part of the effort to address criticisms of the exclusive use of normalised length changes in the submitted draft. We see a moderate bias towards underestimating 1950 length from the CESM-driven runs, and towards overestimating from GISS (despite the mean not reflecting this due to the effect of outliers), and a greater range of length changes generated by the BCC-CSM-driven runs.

Distribution of per-glacier differences between modelled and Leclercq-observation length (absolute)

[Figure]

Proposed new figure P4: Plotting the modelled and observed per-glacier trends over the 20th century (including all glaciers which have 68 or more years in the 20th century covered by the Leclercq timeseries, which represents the point where 90% of glaciers are included). This addresses the issue raised of the glaciers being represented only through regional means. The data shows that the magnitude of observed trends on the scale of individual glaciers is not well modelled by OGGM, and that the differences in how well represented glacier changes are between models using different GCM forcings are small compared to the difference between the modelled changes and the observed changes. The less-than-parity regression line slopes for every model suggest that OGGM is likely to underestimate glacier retreat, especially for larger values of observed retreat. A similar plot for normalised trends shows an almost identical picture, but we choose the absolute trends simply because the required axis scales are less impacted by outliers.

Per-glacier 20th century trends: modelled vs observed (absolute, all glaciers)

---

## Author Response (AR1)

**Author's response**

We provide the new manuscript with the responses to the reviewers as they were given at the last submission. Because of the format of these responses as a changelist for the new version of the manuscript, these review responses also serve as the list of alterations made to the manuscript in the latest version. Only one substantive change is made beyond those listed in the review responses: the addition of a set of graphs in the supplementary material (S11-16) which show the spread of the modelled glacier lengths for each region and GCM, as requested in the editor's response to the review responses. A marked-up version of the manuscript was not deemed practical given the fact that so much of the manuscript revision involved significant restructuring of entire sections and large blocks of new material. We hope that this is understandable given the large changes to the manuscript that were necessary based on the reviewers' comments.

---

## Referee Report (RR1)

I agree with the previous reviewers, that the questions that are being addressed in the manuscript are very interesting and fit to the Cryosphere. Significant improvements have been made based to the manuscript based on the previous reviews. Unfortunately the improvements are in my opinion insufficient to recommend publication in its current state. I like the revised introduction and most of it is easy to read, minor comments you will find below. I have however trouble understanding the experimental set-up that was used. The methods need to be clarified, as there are a couple of things that seem inconsistent and/or confusing to me. This is being specified below, together with some other detailed comments on the section. I will keep the remaining part of this review very brief, because based on the method description, I am not certain about what the model set-up is and therefore on how to value the results.

Without a version with marked changes of the manuscript it was very hard to track what improvements have been maybe. However I found some comments that neither have been addressed in the rebuttal letters nor implemented in the revised manuscript. I would recommend the authors to go over the reviews again and check if they addressed all comments. Though this maybe sounds all very negative, I think the changes that are required might be easy to implement and I see potential in the paper.

**2. Data and model description**

2.1 As you're analyzing changes in glacier length, I would recommend to spend a few words on describing the dynamic part of the model in this section. (At it is current state the reader will only find out it is a flowline model in the "Limitation and Extensions" section.)

Line 88: *"OGGM is an open source model of glacier evolution, which couples a surface mass balance (SMB) model based on solid precipitation and temperature with a model of glacier dynamics."*

This sentence makes me wonder, did you force your simulations with solid precipitation? If so why, did you do so? (OGGM is calibrated with total precipitation and computes the solid fraction based on that.) Please clarify the type of precipitation you used, either here or in line 111 for instance.

Line 94: "*using CRU 3.21 temperature and precipitation data for the 20th century (Jones and Harris, 2013)*"

OGGM uses CRU TS v4.01 (Harris et al., 2014) anomalies with respect to the CRU CL (New et al., 2002) in its calibration procedure (see Maussion et al., 2019 Appendix A for the details). In this context I am wondering, to which CRU data are you referring in the section Line 111-115. Is this the by OGGM preprocessed CRU climate data? If so this needs to be clearly indicated. If not, why do you use a different climate than was used in the calibration procedure, when applying the anomaly method to the GCM data?

Line 109: Again be more specific. What type of temperature (reference height temperature?) and precipitation (total/ solid) did you use?

Line 115-116: "*The processing is done within OGGM, as its default setting.*" and Line 119, "*using OGGM's default preprocessing at level 3*" contradict what is being stated in line 108-110. "*The GCMs provide gridded monthly records of temperature and precipitation, which OGGM uses to determine temperature and precipitation at each specific glacier location using locally-calculated lapse rates for each.*"

OGGM uses by default a lapse rate of 6.5 C/km. This makes me wonder, did you use a different lapse rate in your forward runs than was used in the calibration? Additionally I wonder, where did you get the "locally calculated lapse rate" from? (To my knowledge this is not an implemented option in the OGGM function that process the GCM data.)

Line 118-123: It is odd to start the "Glacier observations" section with the model set-up. The remaining part of the this sections is in contrast to the first part of the method section written less messy and more easy to follow.

Line 123: Maybe you could mention the inversion method somewhere.

Fig. 2: Consider adding the names of the regions in the legend.

Fig. 2: I think the word "number" is missing before "of glaciers".

**3. Experimental description**

Section 2 and 3: The order in which the methods are described, was inconvenient for me to read, mostly because parts of the experimental set-up is described spread throughout the "data and model description" section and the other parts are in the "experimental description". I recommend to make section 2 strictly a description of the model and data, or merge the two sections. Either way I would just describe the model set-up that you used in one section, instead of spread over multiple sections with other descriptions in between.

Line 188: Do the spin-up runs with the different forcing results in very different glacier lengths? Why did you decide to start from the different simulations from a different initial state?

Line 190: "*For all runs, a 300-year spinup using annual climate data selected randomly from a 51-year window of 875-925 CE from the same model is performed prior to the run, to allow the glacier to develop from the preprocessed glacier geometry (based on RGI data and therefore representative of the year of the observation used) to a more realistic geometry for the climate near the start of the model run.*"

I find the statement quite strong. What is it based on?

**4-7. Results – Conclusion**

- How sensitive are the results to the year that was picked for normalization (line 148-149)?

- Line 215: Are there glaciers in the Leclercq dataset that are expected to have been retreating throughout the last millennium?

- Fig 3&4. Ref table 1. It is not clear from the table which glaciers are included in the plot. Are the glaciers that have failed during one of the the simulations included in figure 3 and 4?

- I'm wondering, how do your results for Europe differ from those in Goose et al. (2018)?

- Line 347-365: Are you aware that there is a function in OGGM which can merge the glacier flowlines?
([https://docs.oggm.org/en/latest/generated/oggm.workflow.merge_glacier_tasks.html](https://docs.oggm.org/en/latest/generated/oggm.workflow.merge_glacier_tasks.html)) Have you tested the claim that you make in the last part of this section?

- Line 376-378: Consider rewording.

- I noticed that very little references to the literature are being made in the *Discussion* and *Limitations and Extensions* section.

**Minor details:**
- The comment on line 25 (previously line 21), by reviewer 3, hasn't been addressed nor included: *most relevant study from Oerlemans to supportthis statement is Oerlemans (2005)*
- I got lost in the sentence that starts in line 24 and end in line 28. Where does *",however (Zemp et al., 2015; Cogley, 2009),"* refer too?
- Line 33: I think the reference to the website in redundant.
- Line 36: Update the reference to the CRU version that is being used by OGGM v1.1.
- Line 37: I think it would be nice to name the version of the RGI that you used some where. (I know this can be looked up with the reference. However I think it is common practice to name the version some where.)
- Line 46-47: I think the it would be appropriate to use a reference for the timing of the LIA and MWP.
- I would like to repeat the suggestion of reviewer 3, to try to avoid multiple repetitions of brackets. (e.g. line 32 & 118). In addition please remover double brackets (e.g. Line 69).
- e.g. line 208: I would like to suggest to use the names of the regions and not their numbers.

---

## Author Response (AR2)

**Author's response**

We thank the editor and the referees for their comments, and in particular referee 4 for stepping in at a late stage, and for providing informed critiques of our description of OGGM and the experiment setup. We believe this has helped to remove any remaining errors in the technical description of our work.

Below we respond to each of the issues raised, referencing where relevant the line in the updated manuscript where the changes are included (referee/editor comment in bold, author response unbolded).

**2.1 As you're analyzing changes in glacier length, I would recommend to spend a few words on describing the dynamic part of the model in this section. (At it is current state the reader will only find out it is a flowline model in the "Limitation and Extensions" section.)**

Some description of the flowline model has been added (line 186). It is fairly limited as interested parties will want to use OGGM documentation, but lists the most important points and specifically describes the way glacier length increases or decreases.

**Line 88: "OGGM is an open source model of glacier evolution, which couples a surface mass balance (SMB) model based on solid precipitation and temperature with a model of glacier dynamics." This sentence makes me wonder, did you force your simulations with solid precipitation? If so, why, did you do so? (OGGM is calibrated with total precipitation and computes the solid fraction based on that.) Please clarify the type of precipitation you used, either here or in line 111 for instance.**

Temperature and precipitation data usage has been clarified. It is now explicit that OGGM has just temperature and precipitation as climate inputs, and that the processing to use degree months (from temperature) and solid precipitation (from total precipitation) is part of OGGM's internal processing. (lines 91-95)

**Line 94: " using CRU 3.21 temperature and precipitation data for the 20th century (Jones and Harris, 2013) " OGGM uses CRU TS v4.01 (Harris et al., 2014) anomalies with respect to the CRU CL (New et al., 2002) in its calibration procedure (see Maussion et al., 2019 Appendix A for the details). In this context I am wondering, to which CRU data are you referring in the section Line 111-115. Is this the by OGGM preprocessed CRU climate data? If so this needs to be clearly indicated. If not, why do you use a different climate than was used in the calibration procedure, when applying the anomaly method to the GCM data?**

Reference has been changed (lines 36, 93, 171, 178). This was a holdover from outdated information that was probably missed due to the major model/data description changes for the revised version of the paper. Both OGGM's default processing and our runs do use the CRU TS 4.01 dataset.

Note: the reference suggested and the reference used by Maussion et al is actually for the CRU 3.10 paper, but there is a new (more recent than the actual data product) paper on the 4th CRU dataset which should be used for 4.01

**Line 109: Again be more specific. What type of temperature (reference height**

**temperature?) and precipitation (total/ solid) did you use?**

Use of CRU reference height made explicit, and climate variable use overall clarified. (lines 91-95)

**Line 115-116: " The processing is done within OGGM, as its default setting. " and Line 119, " using OGGM's default preprocessing at level 3" contradict what is being stated in line 108-110. " The GCMs provide gridded monthly records of temperature and precipitation, which OGGM uses to determine temperature and precipitation at each specific glacier location using locally-calculated lapse rates for each. " OGGM uses by default a lapse rate of 6.5 C/km. This makes me wonder, did you use a different lapse rate in your forward runs than was used in the calibration? Additionally I wonder, where did you get the "locally calculated lapse rate" from? (To my knowledge this is not an implemented option in the OGGM function that process the GCM data.)**

Temperature lapse rate is corrected (the 6.5 degree default is the actual method used) (line 173). With the reordering of the data description and OGGM function description, this lapse rate application (and the rest of the procedure for processing GCM data internally) is now described under the OGGM section, not the GCM data section.

 Important note: an additional correction to the text is the description of the adjustment to the GCM means. This is actually relative to the 1961-1990 CRU TS 4.01 mean, as is the default, and it is better described as a process of applying anomalies rather than 'correcting means'.

**Line 118-123: It is odd to start the "Glacier observations" section with the model set-up. The remaining part of this sections is in contrast to the first part of the method section written less messy and more easy to follow.**

See the response for 'Section 2 and 3'.

**Line 123: Maybe you could mention the inversion method somewhere.**

We looked at this, and in the context of lines added on OGGM's flowlines and length change process (more critical to our outputs) we would like to avoid going into the inversion process explicitly. There are many internal features of OGGM's processing and it would be preferable to be explicit about those which deal with out chosen inputs (GCM data processing) and chosen outputs (definition of length and length changes). Ultimately, the inversion process and associated ice thickness can be left to OGGM's 'black box' for readers who aren't specifically interested in OGGM's internal workings, and those who are interested are better served by OGGM documentation.

**Fig. 2: Consider adding the names of the regions in the legend.**

Abbreviated region names have been added to the legend (abbreviated so as not to take up too much of the graph's space).

**Fig. 2: I think the word "number" is missing before "of glaciers".**

'Number' added in front of 'of glaciers'.

**Section 2 and 3: The order in which the methods are described, was inconvenient for me to read, mostly because parts of the experimental set-up is described spread throughout the "data and model description" section and the other parts are in the "experimental description". I recommend to make section 2 strictly a description of the model and data, or merge the two sections. Either way I would just describe the model set-up that you used in one section, instead of spread over multiple sections with other descriptions in between.**

Sections have been reordered, with 'data description' first, and the OGGM-specific information moved into 'OGGM and experiment description'. This makes the reading experience smoother with first the data, then OGGM itself, then the specific setup of OGGM for our experiments.

**Line 188: Do the spin-up runs with the different forcing results in very different glacier lengths? Why did you decide to start from the different simulations from a different initial state?**

This comment was a bit hard to process. Hopefully the change in response to the next comment addresses some of the need for additional clarity on the spin-up of the run.

**Line 190: " For all runs, a 300-year spinup using annual climate data selected randomly from a 51-year window of 875-925 CE from the same model is performed prior to the run, to allow the glacier to develop from the preprocessed glacier geometry (based on RGI data and therefore representative of the year of the observation used) to a more realistic geometry for the climate near the start of the model run." I find the statement quite strong. What is it based on?**

Removed the assertion that the initial glacier geometry is 'realistic' and stated more correctly that the spin-up produces a glacier geometry closer to equilibrium for the start-of-run climate, removing the likelihood of unrealistic large adjustments in glacier geometry at the start of the run. Also changed 'model' to 'GCM' here for greater specificity in the random climate selection. (lines 204-209)

**How sensitive are the results to the year that was picked for normalization (line 148-149)?**

There isn't really an opportunity to choose a different normalisation year because 1950 is the year which is guaranteed to be covered by the Leclercq length change timeseries. This is made more explicit in the text. The impact of changing the normalisation year wouldn't really generate anything interesting as the normalisation year just establishes a scaling factor for each entire timeseries rather than changing any of the qualitative aspects of the data display.

**Line 215: Are there glaciers in the Leclercq dataset that are expected to have been retreating throughout the last millennium?**

The wording has been very slightly changed ('is a modelling issue' -¿ 'is most likely a modelling issue') (line 226) to make it less strong, but in general the point should stand. Discussing whether there are Leclercq glaciers that are expected to retreat from 851 onwards requires indepth references to proxy data, since direct observations are not available for these glaciers at

the time. This would risk significantly diluting the key point: that we have provided a cut-off in the timeseries of results compared to the full timeseries modelling to remove modelling artefacts; something that has value even if there are certain glaciers that do show dramatic retreat in the period that has been cut. It is possible to discuss this if it is deemed necessary, but in the main text of the paper it feels like anything added on this point will add bloat and potentially confusion more than it will improve the explanation of the process.

**Fig 3&4. Ref table 1. It is not clear from the table which glaciers are included in the plot. Are the glaciers that have failed during one of the the simulations included in figure 3 and 4?**

The table has been reworked to instead list the total number that contribute to the mean in each region (the same information, but the complement of the set removed due to failures as was shown in the old version) to make it immediately clear how many glaciers are averaged for each line in figures 3 and 4.

**I'm wondering, how do your results for Europe differ from those in Goose et al. (2018)?**

A quick comparison with Goosse et al. (2018) is made at the start of the discussion, noting simply that the results are consistent. (line 300)

**Line 347-365: Are you aware that there is a function in OGGM which can merge the glacier flowlines? Have you tested the claim that you make in the last part of this section?**

It's possible this is not 100% correct, as this is quite an involved part of OGGM processing and not part of the 'usual' method of use (the merging function is not called by other parts of OGGM in the 1.1 code: it must be something a user's script explicitly does), but our understanding of OGGM merging glaciers is as follows. OGGM can be requested to merge glaciers with intersecting downstream lines, but it is assumed this can only be done as a preprocessing step, and not on the fly during a dynamic run. To do this during the dynamic run would be fundamentally problematic, as it would mean changing the set of glacier objects while the dynamic run is iterating over each of those glacier objects. A small adjustment is made to what we say about this, referencing the merging option but making it clear this is not an option to do dynamically during a glacier evolution run.

**Line 376-378: Consider rewording.**

This section has been reworded with more care taken to specify what is referred to and how the below-glacier-terminus part of the flowline is determined (lines 396-400).

**I noticed that very little references to the literature are being made in the Discussion and Limitations and Extensions section.**

Unless there are specifics on what aspects are under-referenced, this is quite difficult to respond to. The discussion and limitations sections refer in detail to the results and the design of OGGM/our experiment respectively, so it seems natural that they would contain fewer references than the introduction that sets the context and the data/experiment descriptions that introduce

the external tools being used. References have been added in response to the other points that make the need for specific referencing clear (the Oerlemans omission, LIA and MCA dates, and glacier response times).

**The comment on line 25 (previously line 21), by reviewer 3, hasn't been addressed nor included: most relevant study from Oerlemans to support this statement is Oerlemans (2005)**

The Oerlemans 2017 reference has been replaced with the Oerlemans 2005 reference, as we agree this seems more relevant (Oerlemans 1986 remains as an additional reference here). (line 24)

**I got lost in the sentence that starts in line 24 and end in line 28. Where does ",however (Zemp et al., 2015; Cogley, 2009)," refer too?**

The 'however' in this sentence was actually unnecessary and its removal should fix the odd positioning of the reference. (line 26)

**Line 33: I think the reference to the website in redundant.**

Website reference removed

**Line 36: Update the reference to the CRU version that is being used by OGGM v1.1.**

CRU referencing has been corrected and made consistent. (see above author comment in response to the comment on line 94)

**Line 37: I think it would be nice to name the version of the RGI that you used some where. (I know this can be looked up with the reference. However I think it is common practice to name the version some where.)**

RGI version correctly referenced at the specified line and other relevant points.

RGI version correctly referenced at the specified line and other relevant points. (line 37, 97)

**Line 46-47: I think the it would be appropriate to use a reference for the timing of the LIA and MWP.**

The Neukom et al. (2019) paper has been referenced again here, as this provides an analysis of the timings for the Little Ice Age and Medieval Climate Anomaly (we have now adopted the term 'Medieval Climate Anomaly' over 'Medieval Warm Period' for consistency with that paper).

**I would like to repeat the suggestion of reviewer 3, to try to avoid multiple repetitions of brackets. (e.g. line 32 & 118). In addition please remover double brackets (e.g. Line 69).**

Double-bracketing removed

**e.g. line 208: I would like to suggest to use the names of the regions and not**

**their numbers**

Changed reference to region name rather than number (line 227), and checked consistency of this approach throughout the paper.

**Further minor points by editor:**

**line 26: the position of the citation seem odd in this sentence, should it not be: '...are relatively sparse (Zemp....), however,...'**

The 'however' in this sentence was actually unnecessary and its removal should fix the odd positioning of the reference. (line 26)

**line 32: maybe delete one of the OGGMs in the brackets here, repetition.**

Superfluous 'OGGM' removed. (line 32)

**line 125/126: something wrong with the formatting of the citation, should be: '...compiled by Leclercq et al (2014).**

Reference reformatted. (line 104)

**lines 137/138: I would first mention 'glacier thickness' here and cite Johannesson et al (1989), yes of course slope, etc...  influence thickness but the simple theory by Jóhannesson et al. (1989), and further developed by Harrison et al. (2001) really puts it down to thickness. I would refer to these original papers here. Thickness is to some degree of course linked to size (length) of a glacier, hence, the delay of large glacier.**
**Jóhannesson, T., Raymond, C., andWaddington, E.: Time-scale for adjustment of glaciers to changes in mass balance, Journal of Glaciology, 35, 355-369, https://doi.org/10.1017/S00221 1989.**
**Harrison, W. D., Elsberg, D. H., Echelmeyer, K. A., and Krimmel, R. M.: On the characterization of glacier response by a single time-scale, Journal of Glaciology, 47, 659-664, https://doi.org/10.3189/172756501781831837, 2001.**

The references requested have been added, and the text has been modified to reflect the focus on thickness as a driver of response time. An additional comment and reference have been added associating, in a generalised way, glacier thickness and glacier extent.

**Fig 7: although the scale (limits) of the x and y axis are identical, a 1:1 line in the plot would help**

Lighter 1:1 ratio dotted line has been added, and described in the figure caption.

[revised manuscript text omitted]